# Intermittent dietary methionine deprivation facilitates tumoral ferroptosis and synergizes with checkpoint blockade

Ying Xue[1,9], Fujia Lu[1,9], Zhenzhen Chang[1], Jing Li[1], Yuan Gao[2], Jie Zhou[1], Ying Luo[3], Yongfeng Lai[1], Siyuan Cao[1], Xiaoxiao Li[1], Yuhan Zhou[1], Yan Li[1], Zheng Tan[1], Xiang Cheng[4], Xiong Li[5], Jing Chen[6] & Weimin Wang [1,7,8] ✉

Dietary methionine interventions are beneficial to apoptosis-inducing chemotherapy and radiotherapy for cancer, while their effects on ferroptosis-targeting therapy and immunotherapy are unknown. Here we show the length of time methionine deprivation affects tumoral ferroptosis differently. Prolonged methionine deprivation prevents glutathione (GSH) depletion from exceeding the death threshold by blocking cation transport regulator homolog 1 (CHAC1) protein synthesis. Whereas, short-term methionine starvation accelerates ferroptosis by stimulating CHAC1 transcription. In vivo, dietary methionine with intermittent but not sustained deprivation augments tumoral ferroptosis. Intermittent methionine deprivation also sensitizes tumor cells against CD8$^+$ T cell-mediated cytotoxicity and synergize checkpoint blockade therapy by CHAC1 upregulation. Clinically, tumor CHAC1 correlates with clinical benefits and improved survival in cancer patients treated with checkpoint blockades. Lastly, the triple combination of methionine intermittent deprivation, system x$_c^-$ inhibitor and PD-1 blockade shows superior antitumor efficacy. Thus, intermittent methionine deprivation is a promising regimen to target ferroptosis and augment cancer immunotherapy.

Methionine is an essential amino acid, and its metabolism and biological function have been explored in the context of aging, metabolic disorders, and cancer[1,2]. As a major component of the one-carbon metabolism, methionine contributes to a variety of metabolic processes, including GSH production, polyamine, and nucleotide synthesis, as well as methyl group donation by producing S-adenosylmethionine (SAM)[3,4]. Methionine is also the first amino acid incorporated into the peptide chain, and thus its availability tightly controls protein synthesis[5,6]. Numerous studies have indicated that many types of cancer cells are dependent on the exogenous source of methionine relative to normal cells[3,7]. Considering the major source of methionine is food, dietary methionine intervention has been an attractive strategy for exploring its therapeutic potential, especially for cancer. Methionine-restricted diet has been shown to suppress tumor growth and sensitize tumors to apoptosis-inducing chemotherapy and radiation therapy in various cancer types and mouse models[8–11]. However,

[1]Department of Immunology, School of Basic Medicine, Tongji Medical College, Huazhong University of Science and Technology, Wuhan, China. [2]Department of Oncology, Tongji Hospital, Tongji Medical College, Huazhong University of Science and Technology, Wuhan, China. [3]Department of Laboratory Medicine, Tongji Hospital, Tongji Medical College, Huazhong University of Science and Technology, Wuhan, China. [4]Department of Cardiology, Union Hospital, Tongji Medical College, Huazhong University of Science and Technology, Wuhan, China. [5]Department of Gynecology & Obstetrics, the Central Hospital of Wuhan, Tongji Medical College, Huazhong University of Science and Technology, Wuhan, China. [6]Cancer Center, Union Hospital, Tongji Medical College, Huazhong University of Science and Technology, Wuhan, China. [7]Cell Architecture Research Institute, Huazhong University of Science and Technology, Wuhan, China. [8]Key Laboratory of Organ Transplantation, Ministry of Education; NHC Key Laboratory of Organ Transplantation; Key Laboratory of Organ Transplantation, Chinese Academy of Medical Sciences, Wuhan, China. [9]These authors contributed equally: Ying Xue, Fujia Lu. ✉e-mail: weiminw@hust.edu.cn

the effects of dietary methionine intervention on tumor ferroptosis and cancer immunotherapy are not clear.

Ferroptosis is a non-apoptotic form of cell death caused by the aberrant accumulation of lethal lipid peroxides on cell membrane[12,13]. Normally, lipid peroxides are reduced by glutathione peroxidase 4 (GPX4) using GSH as an indispensable substrate, leading to ferroptosis prevention[14,15]. Inactivation of GPX4 or depletion of GSH is sufficient to cause ferroptosis in various tissues and cells[16]. Cystine, transported through system $x_c^-$, contributes to most of the intracellular cysteine, which is the limiting substrate for GSH synthesis[17]. Deprivation of extracellular cystine or blocking its uptake using system $x_c^-$ inhibitors can cause intracellular GSH depletion and ferroptosis onset[18,19]. Methionine, another source of cysteine generated from the transsulfuration pathway, also contributes to GSH synthesis[20]. Deprivation of methionine or inhibition of the transsulfuration pathway has been shown to promote ferroptosis in cultured tumor cells[21,22]. On the contrary, recent studies indicate that methionine deprivation can suppress ferroptosis in some tumor cell lines[19,23], but the mechanism of action is still unclear.

CHAC1 was discovered as a mammalian proapoptotic factor[24] and later was identified as γ-glutamyl cyclotransferases acting specifically on glutathione for controlling intracellular redox milieu. Over-expression of mammalian CHAC1 in yeast cells leads to increased oxidation and the activation of both TRP-like and L-type calcium channels[25]. In zebrafish, CHAC1 can also regulate its development by changing GSH redox potential and the calcium signaling[26]. Recent studies found that CHAC1 activation might also promote cancer cell death, including ferroptosis or necroptosis[27,28]. And CHAC1 transcription during ferroptosis induction has also been reported to accompany with eIF2α/ATF4-mediated ER stress[27], however, its regulation at the translational level remains elusive.

Recently, the interplay between tumoral ferroptosis and cancer immunity has been revealed gradually. Immune cells such as CD8+ T cells have the ability to sensitize tumor cell ferroptosis through secretion of IFNγ, which reprograms the metabolism of amino acid or fatty acid in tumor cells[29–31]. On the other hand, ferroptosis is considered an immunogenic cell death and is able to promote dendritic cell maturation and activate T cell-mediated adaptive immunity[32–34]. Regardless of the mode of action, synergized antitumor activity has been observed across various mouse tumor models under the combined treatment of checkpoint blockade and ferroptosis targeting agents, including small molecules[30,34,35], therapeutic enzymes[31], and nanoparticles[36,37].

In this study, we unexpectedly found simultaneous methionine deprivation suppressed ferroptosis induced by blockage of cystine uptake in various tumor cell lines and in tumor-bearing mice. By contrast, short-term methionine starvation promotes ferroptosis in response to cystine deprivation and synergizes with system $x_c^-$ inhibitor to induce tumoral ferroptosis in vivo. We identified that CHAC1 induced by cystine deprivation was required to achieve substantial GSH depletion and to ensure ferroptosis onset. We also explored how CHAC1 expression is differently regulated by long-term or short-term methionine deprivation in tumor cells. We find that without using synthetic agents, simply dietary methionine intermittent deprivation has an antitumor activity by sensitizing tumor to ferroptosis and can synergize with PD-1 blockade to inhibit tumor progression and enhance T cell-mediated antitumor immune response.

## Results

### Methionine is required for ferroptosis onset induced by blockage of cystine uptake in vitro and in vivo

Extracellular cysteine and methionine both contribute to the synthesis of intracellular GSH, whose depletion causes ferroptosis[20,38]. Cystine deprivation from the culture medium induced significant cell death (Fig. 1a, b) and increased lipid peroxidation in human fibrosarcoma HT-1080 cells (Fig. 1c). Although methionine deprivation alone failed to induce ferroptosis, we reasoned that it might further enhance cystine deprivation-induced ferroptosis. However, surprisingly we found simultaneous methionine deprivation blocked cystine deprivation-induced cell death (Fig. 1a, b) and lipid peroxidation (Fig. 1c). Erastin, a classical ferroptosis inducer, exerts as an inhibitor of system $x_c^-$ to block cystine uptake and cause GSH depletion[12]. We found that methionine deprivation also inhibited erastin-induced cell death (Fig. 1d). RSL3, another ferroptosis inducer, works by directly blocking GPX4 enzymatic activity. We found that methionine deprivation failed to inhibit RSL3-induced ferroptosis in HT-1080 cells (Supplementary Fig. 1a). To further manipulate cystine uptake genetically, we generated SLC7A11 deficient (SLC7A11$^{-/-}$) HT-1080 cells whose survival are dependent on the supplementation of β-mercaptoethanol (β-ME), which converts extracellular cystine into cysteine. β-ME withdrawal induced significant cell death of SLC7A11$^{-/-}$ cells, which was inhibited under methionine co-withdrawal (Supplementary Fig. 1b). In a variety of human tumor cells, including renal carcinoma cell lines OS-RC-2 (Fig. 1e, f), Caki-1 (Supplementary Fig. 1c), 786-O (Supplementary Fig. 1d), and ACHN (Supplementary Fig. 1e), hepatocarcinoma cell lines SNU-182 (Supplementary Fig. 1f) and SNU-387(Supplementary Fig. 1g), and colorectal adenocarcinoma cell line HT29 (Supplementary Fig. 1h), cystine deprivation were able to induce significant ferroptosis, which was attenuated by simultaneous methionine deprivation. Similarly, simultaneous methionine deprivation could block cystine deprivation-induced ferroptosis in multiple in vitro cultured murine cell lines, including Hepa1–6 (Fig. 1g, h), MC38 (Supplementary Fig. 1i), PancO2 (Supplementary Fig. 1j), B16F10 (Supplementary Fig. 1k), and NIH-3T3 (Supplementary Fig. 1l).

To further explore the effect of methionine deprivation on tumor ferroptosis in vivo, we utilized a methionine-free diet in comparison with a control diet (Supplementary Table 1) and assessed their effects on ferroptosis induced by imidazole ketone erastin (IKE), an in vivo metabolically stable inhibitor of system $x_c^-$[39]. Hepa1–6 cells were sensitive to IKE-induced ferroptosis (Fig. 1i) and lipid peroxidation (Fig. 1j), which were both inhibited by simultaneous methionine deprivation when cultured in vitro. We subcutaneously inoculated Hepa1–6 cells into C57BL/6 mice that were then subjected to the control or methionine-free diet followed by IKE administration. Meantime, a ferroptosis inhibitor liproxstatin-1 was co-administrated to ensure the ferroptosis occurrence in vivo along tumor regression (Fig. 1k). In mice received a control diet, we found IKE treatment efficiently reduced tumor growth, and this effect was completely blocked by liproxstatin-1 (Fig. 1l and Supplementary Fig. 1m, n). However, in mice receiving a methionine-free diet, IKE treatment failed to inhibit tumor growth (Fig. 1m and Supplementary Fig. 1m, n). These effects were further confirmed by measurement of tumors weight at the end of the experiment (Supplementary Fig. 1m). Consistent with previous reports[40], we observed that a methionine-free diet expectedly decreased mice body weight (Supplementary Fig. 1o). We then quantified lipid peroxidation in Hepa1–6 tumor tissue by performing malondialdehyde (MDA) assay (Fig. 1n) and 4-hydroxynonenal (4HNE) immunohistochemistry staining (Fig. 1o). IKE treatment significantly increased tumoral lipid peroxidation in mice received control diet, and these effects were attenuated by dietary methionine deprivation (Fig. 1n, o). We also quantified the expressions of a ferroptosis marker gene PTGS2 and NRF2 downstream genes HMOX1 and TFRC in tumor tissues and found their expressions were all elevated by IKE treatment in mice received a control diet, but not in mice receiving methionine-free diet (Fig. 1p). In addition, the ratio of NAD/NADH in tumor tissues was significantly increased by IKE treatment and was attenuated by liproxstatin-1 co-treatment (Supplementary Fig. 1p).

Therefore, we conclude that methionine is required for ferroptosis execution in response to the blockade of cystine uptake.

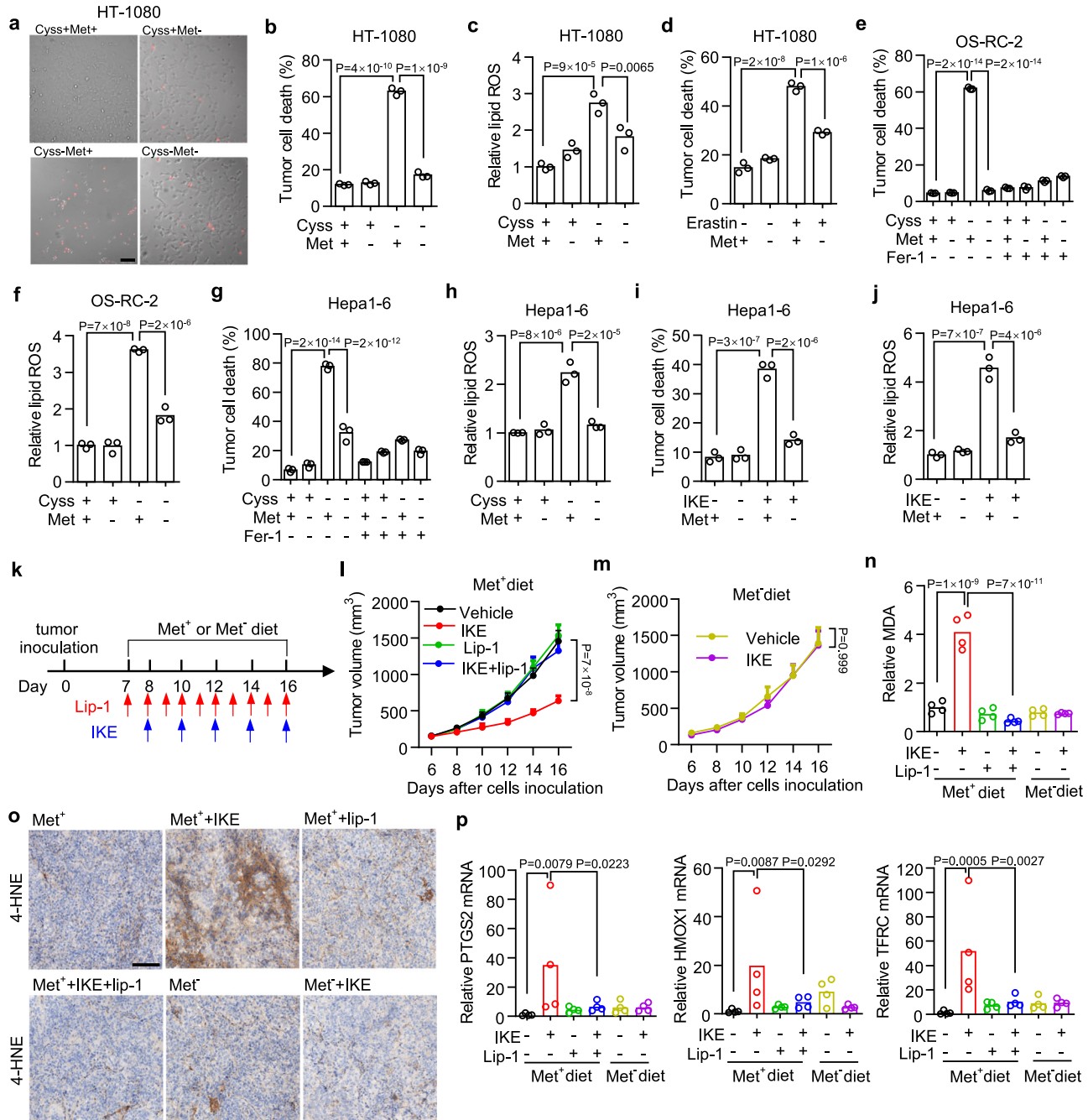

**Fig. 1 | Prolonged methionine deprivation suppresses ferroptosis in response to cystine deprivation. a** Representative images show cell death of HT-1080 cells cultured in a medium that is deficient for cystine (Cyss−) or methionine (Met−) or both of them (Cyss−Met−) for 16 h. Dead cells are visualized by PI staining. **b, c** Cell death (**b**) or relative lipid ROS (**c**) in HT-1080 cells treated with cystine or methionine withdrawal or their co-withdrawal for 20 h (**b**) or 10 h (**c**). **d** Cell death of HT-1080 cells cultured in methionine-free medium and treated with erastin (5 μM) for 24 h. **e, f** Cell death (**e**) or relative lipid ROS (**f**) in OS-RC-2 cells treated with cystine withdrawal plus methionine co-withdrawal in the presence of ferrostatin-1 (Fer-1, 10 μM) for 20 h (**e**) or 8 h (**f**). **g, h** Cell death (**g**) or relative lipid ROS (**h**) in Hepa1−6 cells treated with cystine withdrawal plus methionine co-withdrawal in the presence of Fer-1 (10 μM) for 24 h (**g**) or 12 h (**h**). **i, j** Cell death (**i**) or relative lipid ROS (**j**) in Hepa1−6 cells treated with IKE (1 μM) plus methionine co-withdrawal for

30 h (**i**) or 14 h (**j**). **k–p** Effect of prolonged dietary methionine deprivation on IKE-induced tumoral ferroptosis. Experimental design of Hepa1−6 tumor in C57BL/6 mice (**k**). Tumor-bearing mice were fed with a control (Met+) or methionine-free (Met−) diet and treated with vehicle control, IKE (40 mg/kg), liproxstatin-1 (Lip-1, 10 mg/kg), or their combination. Tumor volume was monitored over time (**l, m**). MDA contents in isolated tumor tissues were quantified (**n**). Images of immuno-histochemistry for 4HNE in tumor tissue slides were presented; scale bars,100 μm (**o**). mRNA levels of PTGS2, HMOX1, and TFRC in isolated tumor tissues were quantified (**p**). $n = 3$ biological replicates and $P$ values are determined using one-way ANOVA (**b–j**); $n = 9$ or 10 mice per group, and data are presented as mean ± s.e.m. and $P$ values are determined using two-way ANOVA (**l, m**); $n = 4$ tumors and $P$ values are determined by one-way ANOVA (**n, p**). Source data are provided as a Source Data file.

Prolonged methionine deprivation suppresses ferroptosis induced by cystine depletion or system $x_c^-$ inhibition both in vitro and in vivo.

## Prolonged methionine deprivation prevents GSH depletion from exceeding the ferroptosis threshold

To explore the mechanism of how methionine metabolism interplay with cystine metabolism, we performed a metabolomics study using a liquid chromatograph mass spectrometer (LC–MS) in HT-1080 cells under cystine or methionine deprivation or their co-deprivation. Total of 148 metabolites belonging to 32 classes were identified from the samples of whole-cell lysates (Supplementary Fig. 2a and Supplementary Data 1). We then focused on the metabolites that were significantly down or upregulated by cystine deprivation, while this dysregulation was diminished by cystine and methionine co-deprivation (Fig. 2a). GSH, the key metabolite regulating ferroptosis, was significantly reduced upon cystine deprivation (Fig. 2b). And its reduction was partially reversed by methionine co-deprivation, although methionine deprivation alone also decreased GSH content (Fig. 2b). This observation was further confirmed by quantifying the intracellular GSH existing in both reduced and oxidized (glutathione disulfide, GSSG) forms[41], using a luminescent-based assay. We found cystine deprivation caused the depletions of both reduced GSH (Supplementary Fig. 2b) and oxidized GSSG (Supplementary Fig. 2c), which were both recovered by simultaneous methionine deprivation (Supplementary Fig. 2b, c). Erastin treatment also resulted in substantial GSH depletion, which is reversed by methionine co-deprivation (Fig. 2c). Similarly, in cultured Hep3B (Supplementary Fig. 2d), 786-O (Supplementary Fig. 2e) and Hepa1-6 (Fig. 2d) cells, cystine deprivation caused substantial intracellular GSH depletion, which was significantly rescued by simultaneous methionine deprivation. In subcutaneous inoculated Hepa1–6 tumors, IKE treatment significantly reduced tumor tissue GSH content in mice receiving a control diet but not in mice received a methionine-free diet (Fig. 2e). These data suggest that prolonged methionine deprivation reverses GSH depletion caused by ferroptosis inducers.

It is believed that cystine deprivation causes GSH depletion mainly by blocking its de novo synthesis, while methionine deprivation can also reduce GSH since it contributes to GSH synthesis through the transsulfuration pathway (Fig. 2f). To trace the fate of cystine and methionine in GSH synthesis, we incubated HT-1080 cells in the medium containing both 200 μM $^{13}$C-labeled cystine and $^{34}$S-labeled methionine for 32 h to reach steady state. Following LC-MS detection and quantification, we found that almost all cysteine and GSH were labeled by $^{13}$C, whereas fewer than 3% of $^{34}$S-labeled cysteine and GSH were detected (Fig. 2g), suggesting that intracellular GSH are mainly from cystine/cysteine-mediated synthesis, but not methionine-mediated transsulfuration pathway, at least in HT-1080 cells. More interestingly, we assessed the GSH degradation by quantifying isotype-labeled cysteine and GSH upon cystine deprivation and found that cystine deprivation caused depletion of both $^{13}$C-labeled cysteine and $^{13}$C-labeled GSH (Fig. 2h). However, cystine deprivation also decreased $^{34}$S-labeled GSH without affecting its precursor $^{34}$S-labeled cysteine (Fig. 2i). These data indicate that cystine deprivation-caused GSH depletion is contributed by promotion of GSH degradation.

We then tested whether a partial recovery of intracellular GSH is sufficient to block ferroptosis by first determining the GSH threshold that is needed to prevent ferroptosis onset. We quantified the intracellular GSH level and matched cell death along the time upon cystine deprivation in both HT-1080 (Fig. 2j) and Hepa1–6 cells (Supplementary Fig. 2f). It was not until the intracellular GSH declined to a certain level (≤6% in HT-1080 or ≤17% in Hepa1–6) that abundant cell death occurred. We then quantified the intracellular GSH concentration after supplementation of exogenous GSH and the corresponding cell death in response to cystine deprivation. In both HT-1080 and Hepa1–6 cells, although supplementation of exogenous GSH only caused a slight recovery of intracellular GSH, it could fully rescue the cell death induced by cystine deprivation (Fig. 2k, l). These results suggest that there is a threshold of GSH inside the cells to mediate ferroptosis resistance, and preventing GSH depletion from exceeding this threshold is sufficient to block ferroptosis, as prolonged methionine deprivation does.

## CHAC1 induction accelerates GSH degradation and ensures ferroptosis onset

Cystine deprivation-induced intracellular GSH depletion is promoted by GSH degradation. Since methionine rarely contributes to GSH synthesis in our system, we speculate that the prevention of GSH depletion from simultaneous methionine deprivation may be caused by the interruption of GSH degradation pathways. It has been previously known that certain de novo synthesized proteins were required for the execution of erastin-induced ferroptosis since a gene transcription inhibitor Actinomycin D (ActD) and a protein synthesis inhibitor cycloheximide (CHX) could both block the ferroptosis[12,42]. We observed the similar inhibitory effects of ActD and CHX on cystine deprivation-induced ferroptosis in our cells (Supplementary Fig. 3a). We thus speculated that certain newly synthesized proteins in response to cystine deprivation were associated with GSH metabolism and would facilitate ferroptosis. To further identify these protein-coding genes, we analyzed the public dataset and found the genes up-regulated by both cystine deprivation[43] and erastin treatment[18] and then cross-referenced them against genes regulating GSH metabolism. SLC7A11 and CHAC1 are known to regulate intracellular GSH abundance and are induced by both cystine deprivation and erastin treatment (Fig. 3a). SLC7A11 composes the system $x_c^-$ to supply cystine for GSH synthesis, while CHAC1 is an enzyme to degrade intracellular GSH into 5-oxoproline (5-OH) and Cys–Gly dipeptide[27,44]. Since cystine deprivation causes GSH degradation, we focused on CHAC1 and confirmed the immediate up-regulation of CHAC1 mRNA upon cystine deprivation or IKE treatment in HT-1080 cells (Fig. 3b). CHAC1 protein level was also increased rapidly after cystine deprivation or IKE treatment in the same cells (Fig. 3c).

To further access the contribution of CHAC1 on cystine deprivation-induced GSH depletion and ferroptosis, we generated CHAC1 deficient HT-1080 cells using the CRISPR technique by transduction of guide RNA targeting CHAC1 (Supplementary Fig. 3b). In cultured control cells expressing scramble guide RNA, cystine withdrawal caused significant reductions of total GSH (Fig. 3d), reduced GSH and oxidized GSSG (Supplementary Fig. 3c), as well as cell death (Fig. 3e), which were all attenuated in CHAC1 deficient cells. Similarly, CHAC1 deficient cells also became resistant to erastin-induced cell death (Fig. 3e). In addition, we knocked down CHAC1 expression in HT-1080 cells by three different short hairpin RNAs (shRNAs), which were all able to decrease CHAC1 protein expression (Fig. 3f), and to reduce cell death induced by cystine deprivation or erastin treatment (Fig. 3g). Consistent with human cells, we manipulated murine CHAC1 by transduction of two different guide RNAs in multiple mouse tumor cell lines including Hepa1-6 (Supplementary Fig. 3d), B16F10 (Supplementary Fig. 3e), MC38 (Supplementary Fig. 3f) and PanO2 (Supplementary Fig. 3g). Guide RNAs could efficiently reduce CHAC1 expression and attenuate cell death induced by cystine deprivation, erastin or IKE treatment (Fig. 3h and Supplementary Figs. 3e–g). These data suggest that cystine deprivation rapidly induces de novo synthesis of CHAC1, which facilities intracellular GSH depletion and ensures ferroptosis onset.

In addition, we dissected the molecular mechanism of how CHAC1 was stimulated by cystine deprivation or erastin treatment during ferroptosis. The deprivation of amino acids, including cystine, could activate endoplasmic reticulum (ER) stress through the eIF2α/ATF4 pathway to initiate the transcription of downstream genes[18,45,46]. We found cystine deprivation (Fig. 3i) or IKE treatment (Supplementary

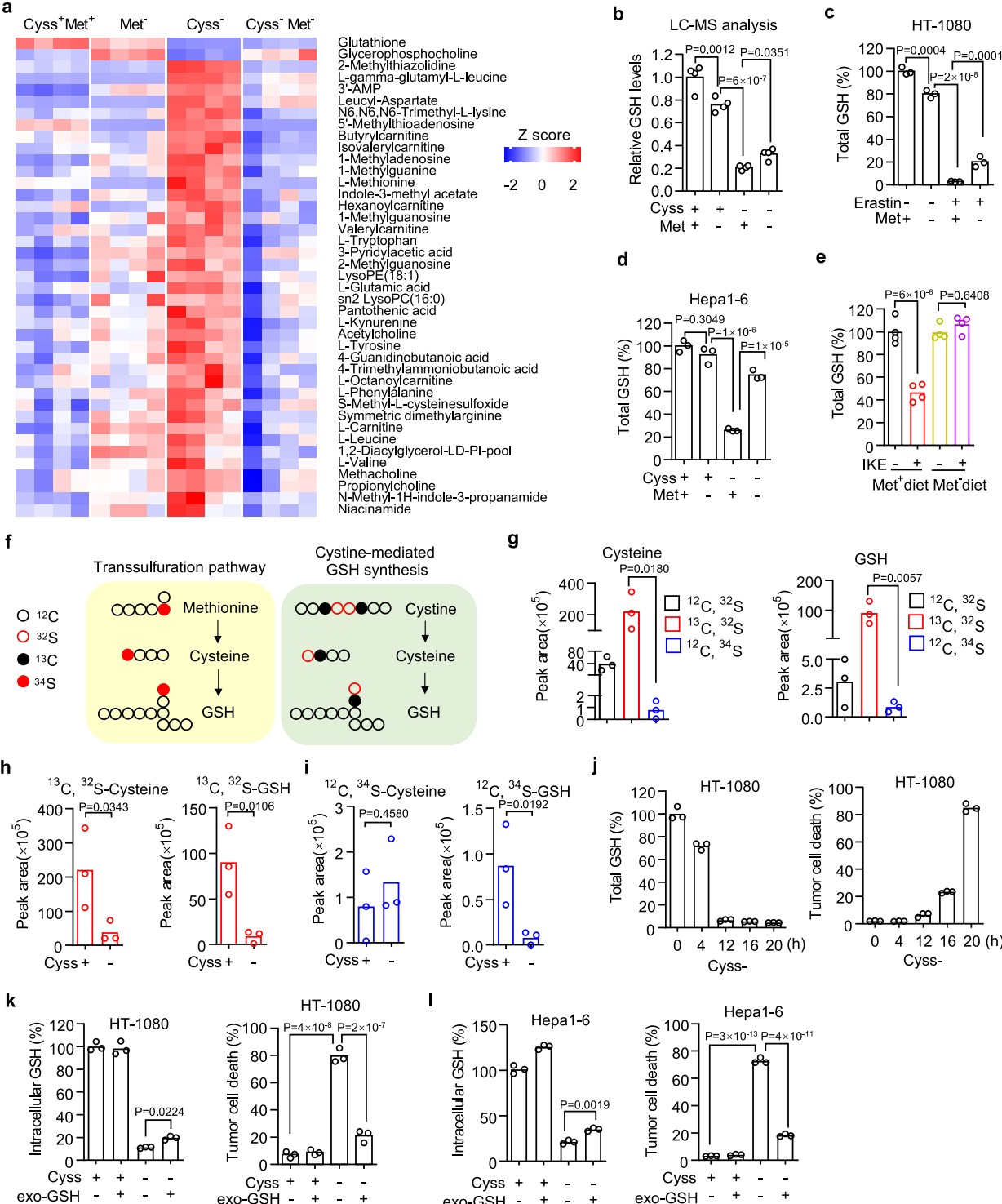

**Fig. 2 | Prolonged methionine deprivation relieves substantial GSH depletion caused by cystine deprivation. a** Heatmap of metabolites that are significantly dysregulated by cystine deprivation and restored by methionine co-deprivation in HT-1080 cells. $n = 4$ biological replicates. **b** LC–MS quantification of GSH abundance in HT-1080 cells treated with cystine or methionine deprivation or their co-deprivation for 8 h. **c** Total GSH content in HT-1080 cells cultured in methionine-free medium and treated with erastin (5 μM) for 9 h. **d** Total GSH content in Hepa1–6 cells cultured in cystine or methionine free or their double-free medium for 10 h. **e** Total GSH abundance in Hepa1–6 tumor tissues from animals treated with control or methionine-free diet plus IKE (40 mg/kg). **f** Schematic of the transsulfuration pathway and cystine-mediated GSH synthesis pathway. Isotype tracing was performed using [13]C-cystine and [34]S-methionine. **g** Abundance of unlabeled and [13]C- or [34]S-labeled cysteine and GSH in HT-1080 cells after incubation with 200 μM [13]C-cystine and [34]S-methionine containing medium for 32 h. **h, i** Abundance of [13]C-labeled cysteine and GSH (**h**) or [34]S-labeled cysteine and GSH (**i**) in HT-1080 cells upon cystine deprivation for 8 h, and cells were pre-incubated with [13]C-cystine and [34]S-methionine for 24 h. **j** Total GSH content and time points matched cell death of HT-1080 cells upon cystine deprivation. **k, l** The rescue effect exogenous GSH (exo-GSH, 200 μM) on intracellular GSH concentration or cell death induced by cystine deprivation for 8 or 20 h in HT-1080 cells (**k**) or for 8 h or 24 h in Hepa1–6 cells (**l**). $n = 4$ biological replicates (**b, e**) or 3 biological replicates (**c, d, g–l**), and $P$ values are determined using one-way ANOVA (**b–e, g–l**). Source data are provided as a Source Data file.

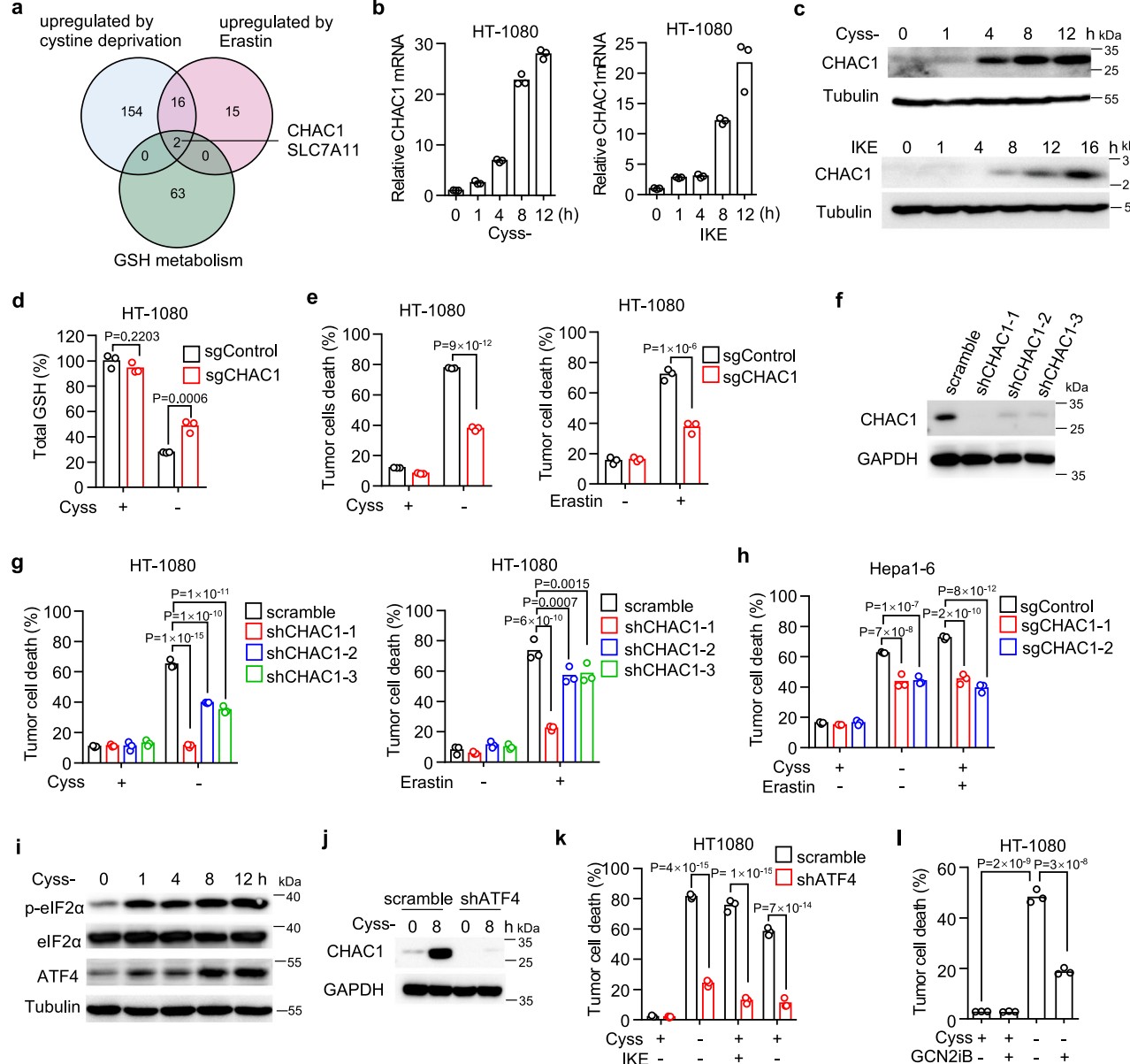

**Fig. 3 | CHAC1 induced by cystine deprivation accelerates GSH depletion and facilitates ferroptosis. a** Venn diagram showing common genes that are upregulated by cystine deprivation and erastin treatment and belong to the geneset of GOBP–glutathione metabolic process. **b** Relative mRNA level of CHAC1 in HT-1080 cells treated with cystine deprivation (left) or 1 μM IKE (right) at different time points. **c** Immunoblot of human CHAC1 in HT-1080 cells treated with cystine deprivation (upper) or IKE (lower) at indicated time points. Tubulin serves as a loading control. **d, e** HT-1080 cells expressing control guide RNA (sgControl) or CHAC1 guide RNA (sgCHAC1) were treated with cystine deprivation or 5 μM erastin. Total GSH content (**d**) was measured at 8 h, and cell death (**e**) was measured at 16 h (for Cyss−) or 20 h (for IKE). **f** Immunoblot of human CHAC1 in HT-1080 cells expressing scramble shRNA or three different CHAC1 shRNA (shCHAC1–1/2/3), and GAPDH serves as loading control. **g** Cell death of HT-1080 cells expressing scramble or CHAC1 shRNA in response to cystine deprivation (left) or 5 μM erastin treatment (right) for 17 h. **h** Cell death of Hepa1–6 cells expressing sgControl or two different sgCHAC1 under 5 μM erastin treatment for 20 h was quantified (**h**). **i** Immunoblots of phosphorylated eIF2α (p-eIF2α), total eIF2α and ATF4 in HT-1080 cells treated by cystine deprivation for indicated time points. Tubulin serves as a loading control. **j** Immunoblot of CHAC1 in scramble or shATF4 expressing HT-1080 cells in response to cystine deprivation for 8 h. GAPDH serves as loading control. **k** Cell death of HT-1080 cells expressing scramble or shATF4 in response to cystine deprivation, IKE (1 μM), or erastin (5 μM) for 18 h. **l** Cell death of HT-1080 cells treated by cystine deprivation plus GCN2 inhibitor (GCN2iB, 1 μM) for 15 h. The experiment was repeated twice with similar results (**c, i**). *n* = 3 biological replicates and P values are determined by two-way ANOVA (**d, e, g–k**) or one-way ANOVA (**l**). Source data are provided as a Source Data file.

Fig. 3h) induced the rapid phosphorylation of eIF2α and upregulation of ATF4 in a time-dependent manner (Fig. 3i and Supplementary Fig. 3h), suggesting that ER stress was activated upon cystine deprivation or IKE treatment. We then manipulated ER stress pathway by knocking down ATF4 with shRNA (Supplementary Fig. 3i). We found that ATF4 knockdown potently reversed CHAC1 upregulation induced by cystine deprivation (Fig. 3j), as well as the ferroptosis in response to

cystine deprivation, erastin or IKE treatment (Fig. 3k). Next, we studied the upstream regulators of eIF2α by focusing on general control nonderepressible-2 kinase (GCN2). GCN2 is known to be activated by direct binding of uncharged tRNAs that accumulate during amino acid starvation, and one of its targets is eIF2α. We found that GCN2 knockdown (Supplementary Fig. 3j) attenuated cystine deprivation-induced ferroptosis (Supplementary Fig. 3k), and similarly, a specific

GCN2 inhibitor GCN2iB could also reverse CHAC1 induction (Supplementary Fig. 3l) and the corresponding cell death (Fig. 3l) in response to cystine deprivation.

Therefore, we conclude that in addition to interrupting GSH synthesis, cystine deprivation or system x$_c^-$ inhibition induces de novo synthesis of CHAC1 through activation of GCN2-eIF2α-ATF4 pathway, leading to exhaustive degradation of GSH and ferroptosis onset.

## Prolonged methionine deprivation blocks the translation of CHAC1

Given simultaneous methionine deprivation inhibited cystine deprivation-induced ferroptosis by interrupting GSH depletion that

was accelerated by CHAC1, we hypothesized that methionine deprivation might directly affect CHAC1 expression induced by cystine deprivation. We then measured CHAC1 expression at both mRNA and protein levels upon single or double deprivation of cystine and methionine. We found that either cystine or methionine single deprivation could upregulate CHAC1 mRNA in a time-dependent manner, and double deprivation also increased CHAC1 mRNA to a similar extent (Fig. 4a). However, at the protein level, although cystine or methionine single deprivation increased CHAC1 expression, their double deprivation failed to further increase CHAC1 and even resulted in a reduction of CHAC1 compared to cystine single deprivation (Fig. 4b). This suggested prolonged methionine deprivation attenuated CHAC1

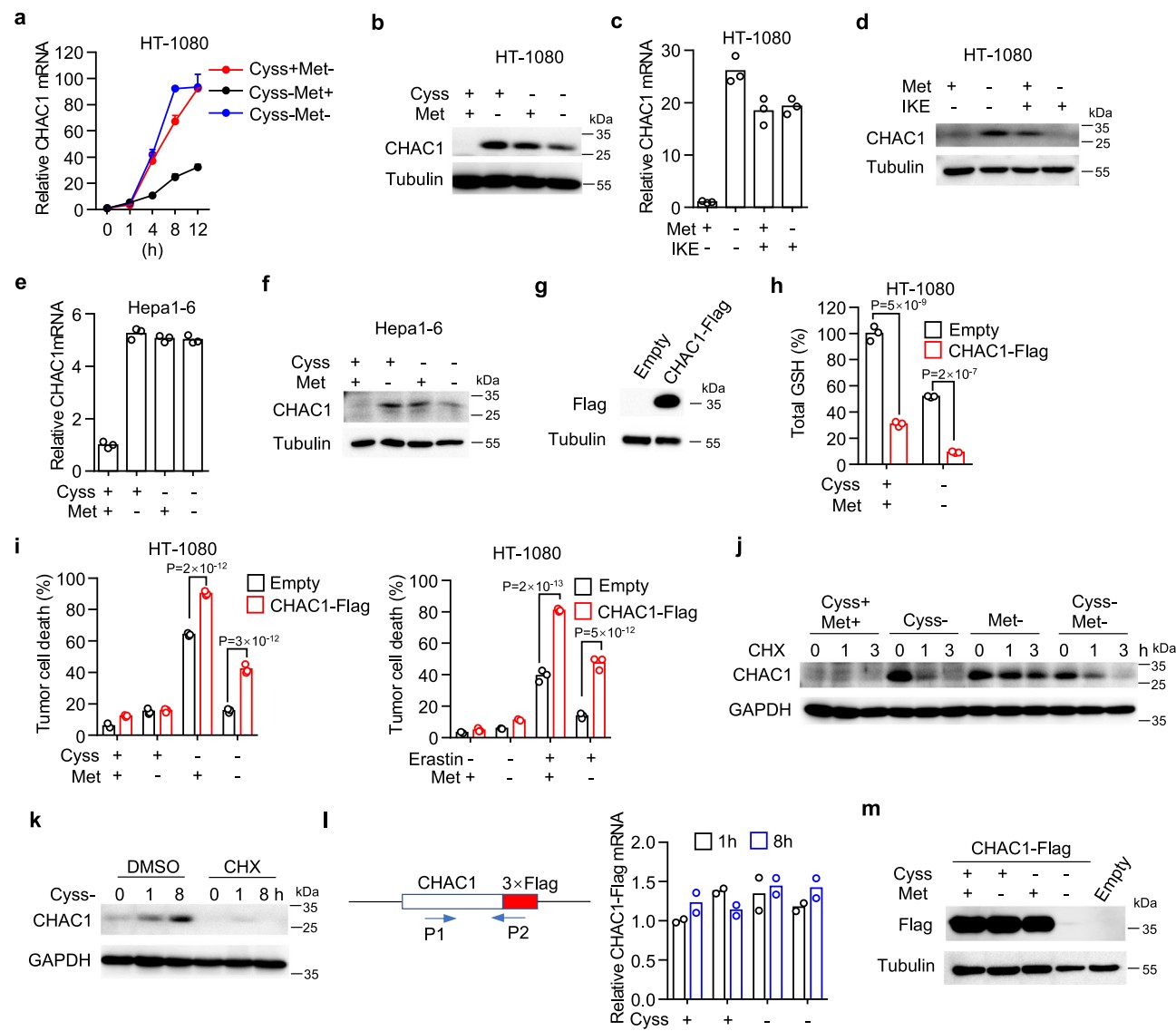

**Fig. 4 | Prolonged methionine deficiency blocks protein synthesis of CHAC1.**
**a**, **b** HT-1080 cells cultured in cystine or methionine free or their double-free medium for indicated times (**a**) or 12 h (**b**). Relative CHAC1 mRNA (**a**) and protein (**b**) are measured. $n = 3$ biological replicates, and data are presented as mean ± SD. **c**, **d** Relative mRNA (**c**) or protein (**d**) of human CHAC1 in HT-1080 cells treated with IKE (1 μM) plus methionine deprivation for 12 h. **e**, **f** Relative mRNA (**e**) or protein (**f**) of mouse CHAC1 in Hepa1–6 cells cultured in cystine or methionine free or their double free medium for 8 h. **g**–**i** HT-1080 cells transduced with control (empty) or CHAC1-Flag expressing lentivirus. Immunoblot of Flag-tagged CHAC1 in these cells (**g**). Cells were treated with cystine withdrawal or 5 μM erastin plus methionine co-withdrawal for 6 h (**h**), 14 h or 16 h (**i**), and total GSH (**h**) or cell death (**i**) was

measured. $n = 3$ biological replicates and P values are determined by two-way ANOVA. **j** Immunoblot of CHAC1 in HT-1080 cells treated with indicated amino acid deprivations for 8 h followed by cycloheximide (CHX) treatment for indicated times. **k** Immunoblot of human CHAC1 in HT-1080 cells treated with cystine deprivation plus CHX (1 μM) for various times. The experiment was repeated twice with similar results. **l**, **m** CHAC1-Flag expressing HT-1080 cells cultured in cystine or methionine free or their double free medium for indicated times (**l**) or 8 h (**m**). Specific primers were designed to amplify CHAC1-Flag mRNA (**l**, left), whose relative expression was quantified (**l**, right). Immunoblot of Flag-tagged CHAC1 under the above conditions (**m**). Source data are provided as a Source Data file.

protein induction caused by cystine deprivation. Similarly, we found that methionine deprivation had no effect on CHAC1 mRNA (Fig. 4c) but reduced CHAC1 protein (Fig. 4d) upregulated by IKE treatment in HT-1080 cells. In mouse Hepa1–6 (Fig. 4e, f), MC38 (Supplementary Figs. 4a, b), and PanO2 (Supplementary Fig. 4c) cells, we also found that simultaneous methionine deprivation attenuated cystine deprivation-induced CHAC1 expression at the protein level (Fig. 4f and Supplementary Figs. 4b, c), but not mRNA level (Fig. 4e and Supplementary Fig. 4a).

To further examine whether CHAC1 reduction contributes to ferroptosis resistance mediated by prolonged methionine deprivation, we cloned Flag-tagged human CHAC1 (CHAC1-Flag) into a lentiviral vector. Surprisingly, we found transient transfection using a higher amount of this plasmid caused significant ferroptosis of HT-1080 cells (Supplementary Fig. 4d), suggesting CHAC1 overexpression is sufficient to initiate ferroptosis without synthetic inducers. We then generated cells stably expressing CHAC1-Flag at a relatively low level (Fig. 4g) and found that forced CHAC1 expression potently decreased the basal level of intracellular total GSH (Fig. 4h), including reduced GSH and oxidized GSSG (Supplementary Fig. 4e). Moreover, forced CHAC1 expression resulted in substantial depletion of total GSH (Fig. 4h), reduced GSH and oxidized GSSG (Supplementary Fig. 4e) even under cystine and methionine double deprivation. In line with GSH depletion, forced CHAC1 expression not only increased ferroptosis induced by cystine deprivation or erastin treatment but also relieved the ferroptosis inhibition mediated by simultaneous methionine deprivation (Fig. 4i). Similarly, in various other tumor cell lines including OS-RC-2 (Supplementary Fig. 4f), 786-O (Supplementary Fig. 4g) and SNU387 (Supplementary Fig. 4h) cells, forced CHAC1 expression was able to enhance ferroptosis induced by cystine deprivation or erastin treatment and recover ferroptosis inhibited by simultaneous methionine deprivation (Supplementary Figs. 4f–h). We also cloned murine CHAC1 cDNA, and its forced expression in B16F10 cells significantly increased ferroptosis induced by cystine deprivation or IKE treatment and also relieved the inhibitory effect of methionine co-deprivation on ferroptosis (Supplementary Fig. 4i).

We next explored the mechanism by which simultaneous methionine deprivation attenuated cystine deprivation-induced CHAC1 expression. Since de novo synthesis of CHAC1 is induced upon cystine deprivation, and methionine co-deprivation attenuated CHAC1 induction at the protein level but not the mRNA level (Fig. 4a). We thus firstly tested whether methionine co-deprivation influenced the stability of CHAC1 protein by the CHX chase assay, which is used to determine the rate of protein degradation when protein synthesis was inhibited. HT-1080 cells were pretreated with amino acids deprivation for 8 h and then followed with CHX treatment for 3 h. We found that cystine deprivation-induced CHAC1 protein rapidly degraded upon CHX treatment, whereas methionine co-deprivation failed to accelerate this degradation (Fig. 4j). We then reasoned that the regulation of methionine co-deprivation on CHAC1 protein might occur at the translational stage. Moreover, methionine is the first amino acid incorporated into the peptide chain, and its deprivation has been reported to strongly inhibit translation initiation and polysome formation[5]. In line with the inhibition of CHX on ferroptosis induced by cystine deprivation (Supplementary Fig. 3a), CHX pre-treatment indeed blocked de novo synthesis of CHAC1 initiated by cystine deprivation (Fig. 4k). Forced CHAC1-Flag expression is driven by human cytomegalovirus (CMV) promoter on the plasmid, therefore exogenous CHAC1-Flag will not be regulated by cystine or methionine deprivation at the mRNA transcriptional level, but should be still regulated at the protein translational level. Indeed, by designing a pair of primers specifically targeting CHAC1-Flag transcript, we found cystine or methionine single deprivation or their double deprivation for 8 h had no effect on CHAC1-Flag mRNA expression (Fig. 4l).

However, cystine and methionine double deprivation strongly inhibited exogenous CHAC1-Flag protein expression, although their single deprivation failed to decrease CHAC1-Flag protein expression over the same time period (Fig. 4m). These results suggest that simultaneous prolonged methionine deprivation suppress cystine deprivation-induced CHAC1 expression at the translational stage, resulting in ferroptosis inhibition.

## Short-term methionine starvation stimulates CHAC1 transcription and promotes ferroptosis

Although simultaneous methionine deprivation in the long term inhibited CHAC1 expression induced by cystine deprivation, we noticed that endogenous CHAC1 mRNA and protein were strongly upregulated within 8 h of single methionine deprivation (Fig. 4a, b). We thus assumed that pre-treatment with short-term methionine starvation (St-Met⁻) might increase CHAC1 expression and then augment ferroptosis induced by subsequent cystine deprivation. Indeed, in various human and mouse tumor cell lines, including HT-1080 (Supplementary Fig. 5a), 786-O (Supplementary Fig. 5b), OS-RC-2 (Supplementary Fig. 5c), and B16F10 (Fig. 5a, b), we found that pre-treatment with methionine deprivation for less than 8 h strongly enhanced ferroptosis in response to cystine deprivation or IKE treatment (Fig. 5a, b and Supplementary Figs. 5a–c).

Moreover, we confirmed that short-term (less than 8 h) methionine starvation increased both CHAC1 mRNA and protein expressions in mouse B16F10 (Fig. 5c) and human HT-1080 cells (Supplementary Figs. 5d, e). Short-term methionine starvation followed by cystine deprivation further increased CHAC1 protein expression (Fig. 5d) and promoted GSH depletion (Fig. 5e). Knockout of CHAC1 with two different guide RNAs could efficiently abolish the cell death induced by short-term methionine starvation in combination with cystine deprivation (Fig. 5f). Similar with the mechanism of CHAC1 induction by cystine deprivation, short-term methionine starvation could also induce the phosphorylation of eIF2α and the expression of ATF4, resulting in CHAC1 upregulation (Supplementary Fig. 5e). Thus, short-term methionine starvation stimulates CHAC1 expression at the transcriptional level via activation of ER stress, resulting in ferroptosis sensitization.

We then tested whether short-term dietary methionine starvation could sensitize tumor cells to IKE-induced ferroptosis in the subcutaneous B16F10 tumor model. Since B16F10 cells are moderately sensitive to IKE-induced ferroptosis in vitro, which is greatly exacerbated by short-term methionine starvation (Fig. 5b). In vivo short-term methionine starvation was achieved by intermittent periods of methionine-free diet feeding and terminated with methionine re-supplementation (Fig. 5g). And the methionine concentration in mouse serum was quantified to verify the effectiveness of dietary manipulation. Methionine level was dramatically reduced by a methionine-free diet within 3 days but was immediately recovered by methionine re-supplementation (Fig. 5h). Meanwhile, IKE treatment alone significantly reduced tumor growth, but surprisingly, dietary methionine intermittent starvation could also reduce tumor growth to a similar extent, and their combination nearly completely stopped tumor progression as shown by tumor volume (Fig. 5i and Supplementary Fig. 5f) and tumor weight (Fig. 5j, k). We then detected the markers of ferroptosis in tumor tissues. The combination of IKE and dietary methionine intermittent starvation dramatically increased MDA content (Fig. 5l), PTGS2, HMOX1, and TFRC mRNA expressions (Fig. 5m) compared with the effect of each treatment alone. In line with this, the combination therapy resulted in significant GSH depletion (Fig. 5n) accompanied by increased CHAC1 mRNA expression (Fig. 5o). These results suggest that short-term methionine starvation sensitizes tumor cells to ferroptosis by stimulating CHAC1 transcription, and dietary methionine intermittent deprivation can sensitize therapeutic efficacy of ferroptosis inducers.

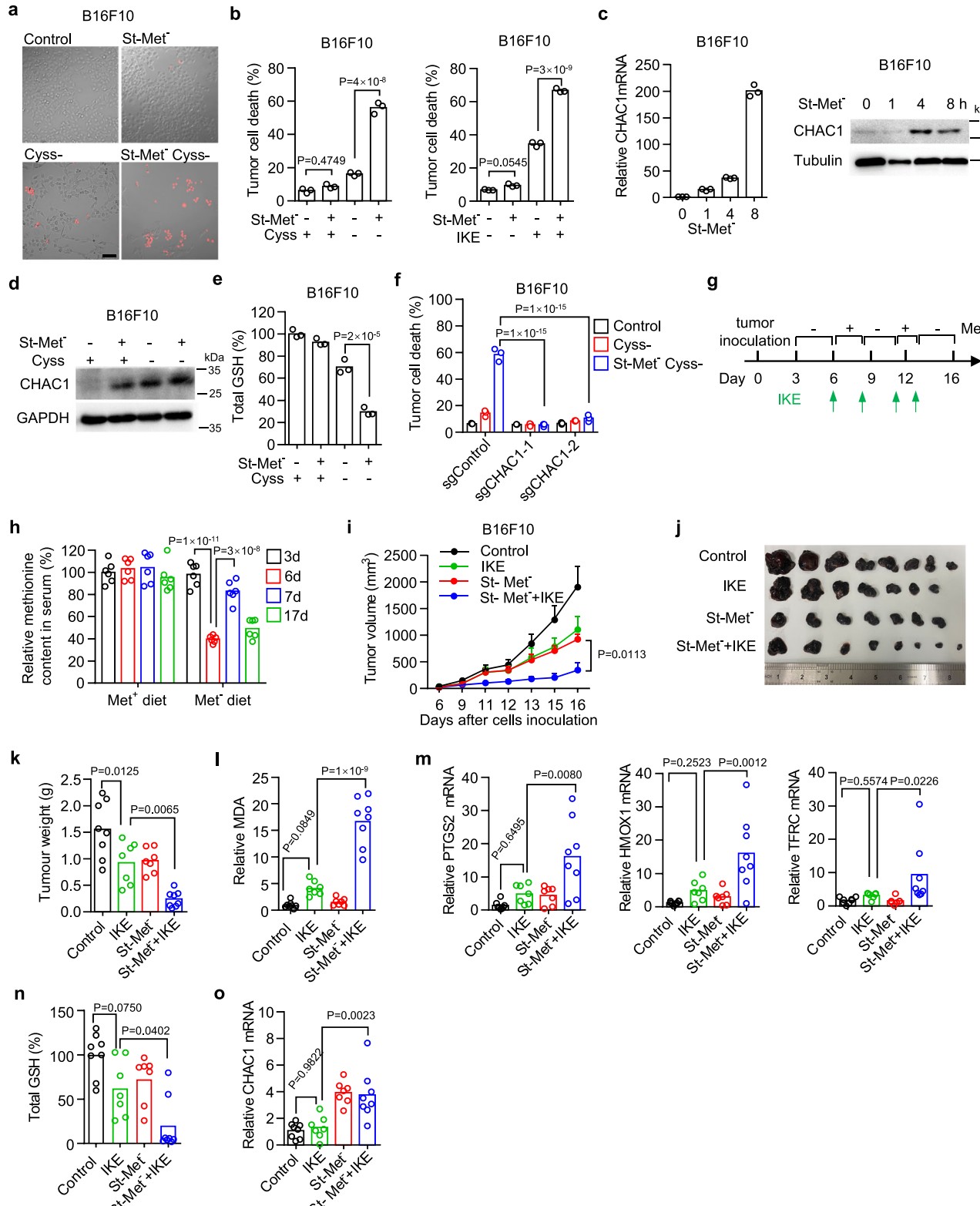

## Short-term methionine starvation sensitizes the tumor against CTL-mediated cytotoxicity and synergizes with checkpoint therapy

CD8+ T cells (CTL) secreted cytokines sensitize tumor cells to ferroptosis, which contributes to its cytotoxicity and in vivo anti-tumor activity mediated by checkpoint blockade[30,31]. Given that short-term methionine starvation promotes tumor cell ferroptosis by induction of CHAC1, we hypothesized that tumor cells undergoing short-term methionine starvation would become more sensitive to CTL-mediated killing. To test this possibility, we established an ovalbumin (OVA) and luciferase (Luc) co-expressing B16F10 cells and co-cultured them with OVA-specific CD8+ T (OT-I) cells, and tumor cell viability was evaluated by quantification of luciferase activity (Fig. 6a). We observed that short-term methionine starvation dramatically increased the

**Fig. 5 | Short-term methionine starvation enhances CHAC1 expression and promotes ferroptosis onset in vitro and in vivo. a, b,** B16F10 cells were pre-starved in methionine-free medium (St-Met⁻) for 8 h, followed by cystine deprivation or 0.5 μM IKE treatment for another 18 h. scale bar,100 μm (**a**), 20 h (**b**, left), or 24 h (**b**, right). Representative images show the induction of cell death that is visualized by PI staining (**a**). Percentages of dead cells were also quantified by flow cytometry (**b**). **c** Relative mRNA (left) or protein expression (right) of mouse CHAC1 in B16F10 cells treated with methionine deprivation for 1–8 h. **d** Immunoblots of CHAC1 in B16F10 cells pre-treated by St-Met⁻ for 4 h followed by cystine deprivation for another 8 h. **e** Total GSH abundance in B16F10 cells pre-treated by St-Met⁻ for 8 h followed by cystine deprivation for another 8 h. **f** Cell death of sgControl or sgCHAC1 expressing B16F10 cells pretreated by St-Met⁻ for 8 h followed by cystine deprivation for another 19 h. **g–o** Effect of dietary methionine intermittent deprivation on IKE-induced tumoral ferroptosis. Experimental design of B16F10 tumor in C57BL/6 mice (**g**). Tumor-bearing mice were fed with a methionine-free (Met⁻) diet or Met⁺ diet and treated with Met re-supplementation (300 mg/kg) plus IKE (40 mg/kg) at indicated days. Methionine concentrations in serum from tail bleeding samples at indicated days were quantified (**h**). Tumor volumes were monitored over time (**i**). Subcutaneous tumors were surgically removed and presented (**j**), and their weights were measured at the endpoint (**k**). MDA content (**l**), PTGS2, HMXO1, and TFRC mRNA expressions (**m**), total GSH abundance (**n**), and CHAC1 mRNA expression (**o**) in isolated tumor tissues were quantified. *n* = 3 biological replicates (**b, e, f**) and *P* values are determined using one-way ANOVA (**b, e**) or two-way ANOVA (**f**); *n* = 6 serum samples and P values are determined by one-way ANOVA (**h**). *n* = 7 (IKE, St-Met⁻) or 8 (Control, St-Met⁻ + IKE) mice per group, and data are presented as mean ± s.e.m. and *P* values are determined by two-way ANOVA (**i**); *n* = 7 (IKE, St-Met⁻) or 8 (Control, St-Met⁻ + IKE) tumors and *P* values are determined by one-way ANOVA (**k–o**). Source data are provided as a Source Data file.

sensitivity of B16F10 cells against CTL-mediated cytotoxicity (Fig. 6a). However, CHAC1 deficiency in B16F10 cells could abolish this sensitization effect mediated by short-term methionine starvation (Fig. 6a). Alternatively, by measuring cell death of CFSE-labeled B16F10 in the co-culture system (Supplementary Fig. 9a), we found that short-term methionine starvation promoted B16F10 cell death in response to OT-I cells, and CHAC1 deficiency also restored this cell death (Fig. 6b). Lipid peroxidation in tumor cells was also quantified to confirm the presence of ferroptosis in above co-culture system (Supplementary Fig. 9b). Short-term methionine starvation synergized with OT-I cells to induce lipid peroxidation in B16F10 cells, which was abolished by CHAC1 deficiency (Fig. 6c). Intriguingly, we found that in the absence of short-term methionine starvation CHAC1 deficient B16F10 cells naturally became resistant to OT-I cells-mediated cytotoxicity (Fig. 6a, b). This CTL resistance caused by CHAC1 deficiency was more obvious in Hepa1−6 cells (Supplementary Fig. 6a, b). Owing to the low basal level of CHAC1 in tumor cells, we speculated that tumoral CHAC1 expression might be elevated by CTL and then conferred to CTL-regulated ferroptosis. Indeed, we found that CHAC1 mRNA expression in tumor cells was increased along the time after incubation with the supernatant from OT-I cells (Fig. 6d). In contrast to CHAC1 knockout, forced expression of CHAC1 resulted in more cell death (Fig. 6e) and enhanced lipid peroxidation (Fig. 6f) of B16F10 cells in response to OT-I cells. Considering that constitutively CHAC1 over-expression might directly affect tumor ferroptosis, we constructed the doxycycline (Doxy)-inducible CHAC1-Flag expressing B16F10 cells (Supplementary Fig. 6c). We observed that Doxy-induced CHAC1 expression could enhance cystine deprivation-induced ferroptosis (Supplementary Fig. 6c), and increase lipid peroxidation and tumor cell death in response to OT-I cells (Supplementary Fig. 6d). These data indicate that short-term methionine starvation cooperates with CD8⁺ T cells to induce tumoral ferroptosis by converging on CHAC1.

Since CD8⁺ T cells are activated during cancer immunotherapy, which has the potential to synergize with ferroptosis-targeting agents[30,31,47,48], we then reasoned that dietary methionine intermittent starvation might cooperate with PD-1 blockade to play synergized anti-tumor activity in vivo. To this end, we treated B16F10-bearing mice with dietary methionine intermittent starvation combined with an anti-PD-1 antibody. Meantime, liproxstatin-1 was co-administrated to test the ferroptosis occurrence in this setting (Fig. 6g). Compared with a single treatment of methionine intermittent starvation or PD-1 blockade, the combination therapy more efficiently inhibited tumor growth (Fig. 6h and Supplementary Fig. 6e). And their therapeutic efficacy was significantly attenuated by liproxstatin-1 as shown by both tumor volume (Fig. 6i and Supplementary Fig. 6e) and weight (Fig. 6j and Supplementary Fig. 6f). Accordingly, CHAC1 and PTGS2 mRNA expressions (Fig. 6k, l) and lipid peroxidation marker 4HNE (Fig. 6m) in tumor tissues were all elevated by the combination therapy compared with PD-1 blockade alone, and these inductions were also reversed by liproxstatin-1 co-treatment (Fig. 6k–m). In addition, we quantified the tumor-infiltrating T cells and their effector functions (Supplementary Fig. 9c). The combination therapy increased the infiltration of CD8⁺ T cells, and the percentages of IFNγ⁺, TNFα⁺ CD8⁺ T cells and the percentages of IFNγ⁺, TNFα⁺ CD4⁺ T cells in the tumor microenvironment (Fig. 6n). While liproxstatin-1 again attenuated the changes of CD8⁺ T cells and cytokine productions (Fig. 6n). Therefore, without utilization of synthetic agents, dietary methionine intermittent starvation can cooperate with PD-1 blockade to enhance antitumor immunity by inducing tumoral ferroptosis.

## Tumor CHAC1 loss causes immunotherapy resistance

We next studied the role of tumor CHAC1 in the anti-tumor effect mediated by the above combination therapy. We inoculated CHAC1 wildtype or deficient B16F10 tumor cells into C57BL/6 mice and treated them with dietary methionine intermittent starvation combined with anti-PD-1 antibody. The combination therapy greatly suppressed tumor growth in CHAC1 wildtype tumors, but its therapeutic efficacy was significantly impaired in CHAC1 deficient tumors (Fig. 7a). In line with this, CHAC1 deficiency in tumor cells impaired the infiltrations of CD8⁺ T cells, and expressions of IFNγ and TNFα in CD8⁺ T cells and CD4⁺ T cells that were all induced by the combination therapy (Fig. 7b). These results suggest that the synergistic anti-tumor activity of combined dietary methionine intermittent starvation and PD-1 blockade is dependent on the expression of CHAC1 in tumor cells.

To further clarify the contribution of CHAC1 on the therapeutic efficacy of mono-immunotherapy, we treated CHAC1 wildtype or deficient B16F10 tumor-bearing mice with anti-PD-1 antibody and found CHAC1 loss in tumor cells caused resistance to anti-PD-1 therapy, revealed by both increased tumor volume (Fig. 7c and Supplementary Fig. 7a) and tumor weight (Fig. 7d, e). We then assessed the potential clinical relevance of CHAC1 in cancer immunotherapy. We examined a publicly available transcriptomic profiles of tumor biopsies from melanoma patients treated with PD-1 blockade or combined PD-1 and CTLA-4 blockade[49]. The expression of CHAC1 at baseline was enriched in the cohort who responded to immunotherapy (complete or partial response) compared with non-responders (stable or progressive disease) (Fig. 7f). Meanwhile, high levels of CHAC1 expression were associated with improved overall survival or progression-free survival in these patients (Fig. 7g). Similarly, the analysis of another cohort from Cancer-Immu algorithm[50,51] suggested that CHAC1 expression was also elevated in melanoma patients who responded to CTLA-4 blockade therapy (Supplementary Fig. 7b), and patients with higher CHAC1 expression had increased overall survival or progression-free survival (Supplementary Fig. 7c). Thus, CHAC1 could be a biomarker for a good response to checkpoint blockade in patients with melanoma. Taken together, these results reveal that tumor CHAC1 affects antitumor immunity in vivo via the regulation of tumor ferroptosis.

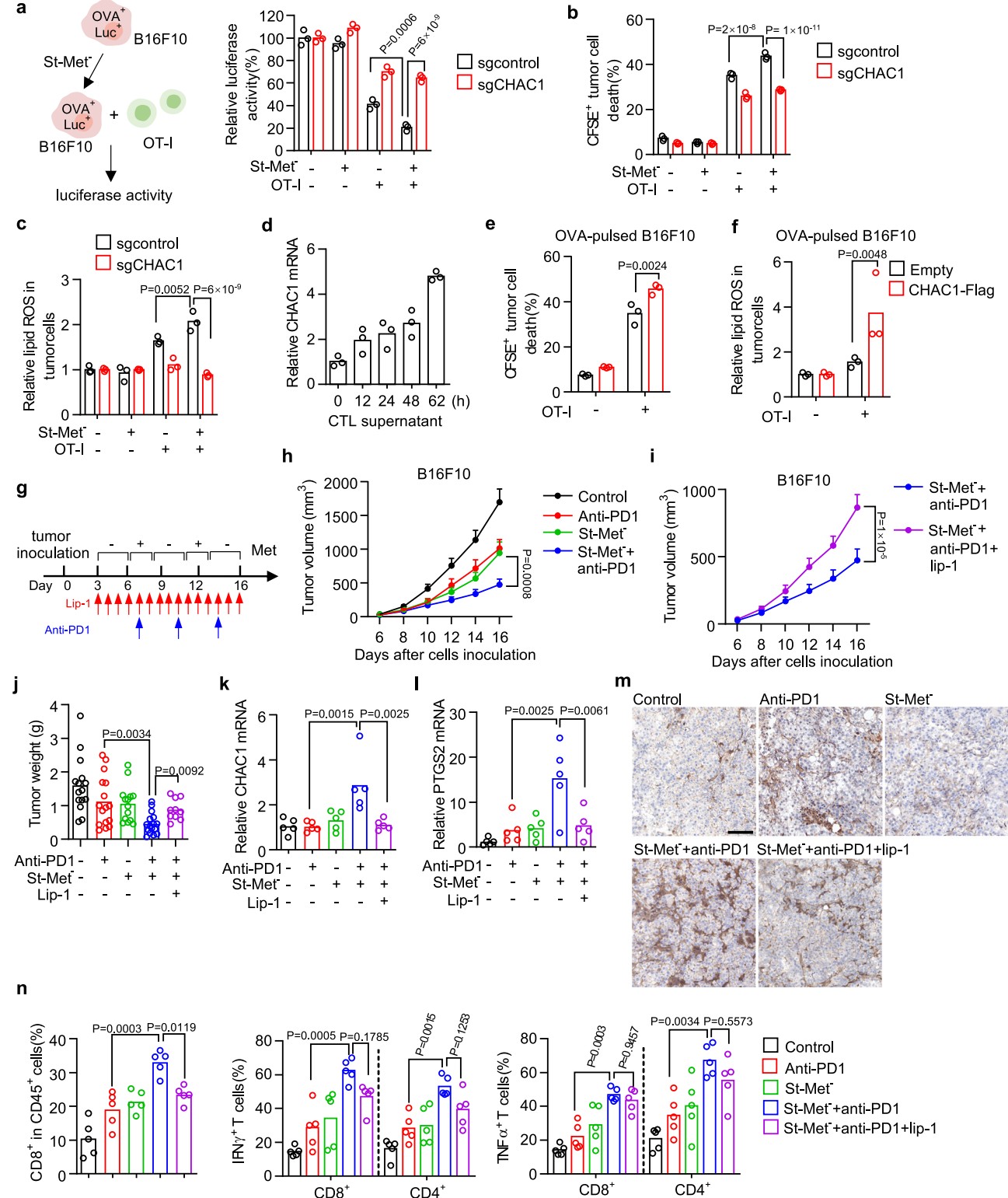

## A triple combination of dietary methionine intermittent starvation, system xc⁻ inhibitor, and PD-1 blockade shows superior antitumor efficacy

We had observed that dietary methionine intermittent starvation could sensitize both ferroptosis-targeting therapy and cancer immunotherapy, thus speculating that the combination of these three treatments would generate the most potent anti-tumor activity. To this end, we performed treatments on subcutaneous B16F10 tumors with only two cycles using dietary methionine intermittent starvation, anti-

PD-1 antibody, and a lower dose of IKE (Fig. 8a). Compared with the control group, a double combination of either IKE plus PD-1 blockade or methionine intermittent starvation plus PD-1 blockade significantly inhibited tumor growth (Fig. 8b, c). While their triple combination further reduced tumor volume in comparison to double combinations (Fig. 8b, c). To solidify the superior antitumor activity of the triple combination, we repeated these treatments by comparing with another combination group of methionine intermittent starvation plus IKE, and found that the triple combination showed more potent

**Fig. 6 | CHAC1 induction by methionine intermittent deprivation sensitizes the tumor against CTL and PD-1 blockade therapy. a–c** Experimental design for co-culture of ovalbumin (OVA⁺) and nano-luciferase (Luc⁺) expressing B16F10 and activated OT-I cells (**a**, left). Control (sgControl) or CHAC1 deficient (sgCHAC1) OVA⁺Luc⁺ B16F10 cells were treated with St-Met⁻ for 8 h and then co-cultured with OT-I cells (tumor: T = 1: 8). Luciferase activity from tumor cells was quantified (**a**, right). Cell death (**b**) or lipid ROS (**c**) of tumor cells in the co-culture system was determined. $n = 3$ biological replicates and $P$ values are determined using two-way ANOVA. **d** Relative mRNA expression of CHAC1 in B16F10 cells treated with supernatant from activated OT-I cells (CTL) for indicated times. **e, f** Control (Empty) or CHAC1-Flag over-expressing B16F10 cells were pulsed with OVA$_{257-264}$ peptide and then co-cultured with OT-I cells for 60 h (**e**) or 36 h (**f**). Cell death (**e**) or lipid ROS level (**f**) in tumor cells from the co-culture system was quantified. **g–j** Effect of

dietary methionine intermittent deprivation on PD-1 blockade-mediated cancer immunotherapy. Experimental design of B16F10 tumor in C57BL/6 mice ($n = 16$ or 10 mice per group) (**g**). Tumor volumes were monitored over time (**h, i**), and tumor weights were measured at the endpoint (**j**). Data were presented as mean ± s.e.m. and $P$ values are determined by two-way ANOVA (**h, i**) and one-way ANOVA (**j**). CHAC1 (**k**) and PTGS2 (**l**) mRNA levels in isolated tumor tissues were quantified ($n = 5$ tumors per group). $P$ values are determined by Kruslal–Wallis with Dunn's multiple comparisons test (**k, l**). Images of immunohistochemistry for 4HNE in tumor tissue slides were presented; scale bars,100 μm (**m**). The percentages of CD8⁺ T cells in CD45⁺ cells and the percentage of cells expressing IFNγ or TNF in CD8⁺ and CD4⁺ T cells were quantified (**n**). $n = 5$ tumors per group, and $P$ values are determined by one-way ANOVA (**n**). Source data are provided as a Source Data file.

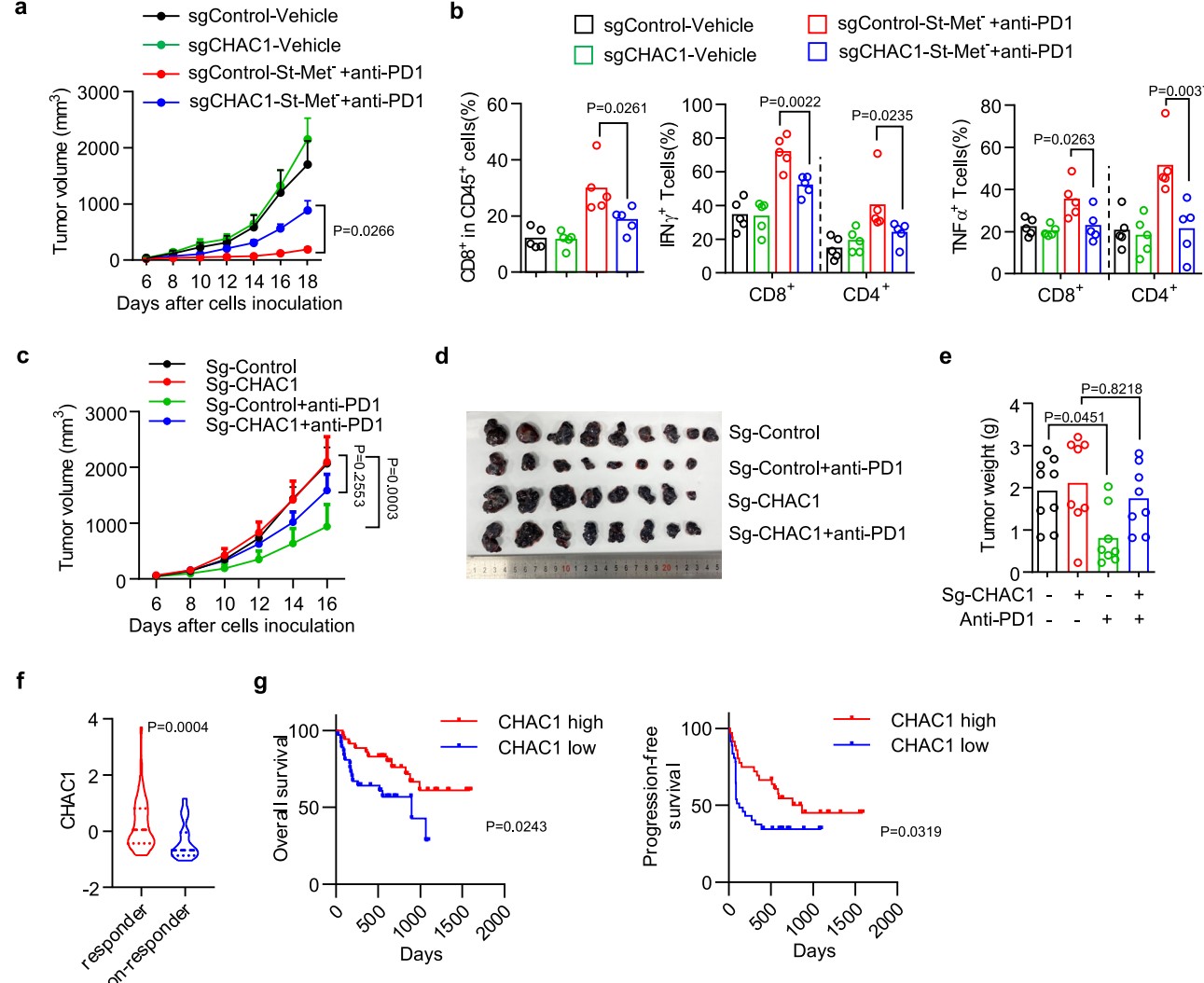

**Fig. 7 | CHAC1 loss in tumor cells impairs antitumor immunity. a, b** Effect of tumoral CHAC1 on combination therapy of St-Met⁻ plus PD-1 blockade. CHAC1 wildtype (sgControl) or deficient (sgCHAC1) B16F10 tumors-bearing mice were treated by dietary methionine intermittent deprivation plus anti-PD-1 antibody. Tumor volumes were monitored over time (**a**). $n = 7$ (sg-Control-vehicle, sg-Control-St-Met⁻+anti-PD1, and sg-CHAC1-St-Met⁻+anti-PD1) or 8 (sg-CHAC1-vehicle) mice per group, data were presented as mean ± s.e.m. and $P$ values are determined by two-way ANOVA (**a**). The percentages of CD8⁺ T cells in CD45⁺ cells and the percentage of cells expressing IFNγ or TNF in CD8⁺ and CD4⁺ T cells (**b**). $n = 5$ tumors per group, $P$ values are determined by one-way ANOVA (**b**). **c–e** Effect of tumoral CHAC1 on PD-1 blockade monotherapy. CHAC1 wildtype (sg-Control) or

deficient (sg-CHAC1) B16F10 tumors bearing mice ($n = 8$ or 9 mice per group) were treated with anti-PD-1 antibody since day 3. Tumor volumes were monitored over time (**c**). Data were presented as mean ± s.e.m. and $P$ values are determined by two-way ANOVA (**c**). Subcutaneous tumors were surgically removed and presented (**d**), and their weights were measured (**e**). **f, g** Violin plot comparing tumoral CHAC1 expression between responders and non-responders of melanoma patients who received anti-PD-1 monotherapy or anti-CTLA-4 plus anti-PD-1 combination therapy[49] (**f**). Kaplan–Meier plots of overall survival (**g**, left) and progression-free survival (**g**, right) for these melanoma patients whose tumors demonstrate high vs. low expression of CHAC1. $P$ values are determined by the two-sided Mann–Whitney test (**f**) or log-rank test (**g**). Source data are provided as a Source Data file.

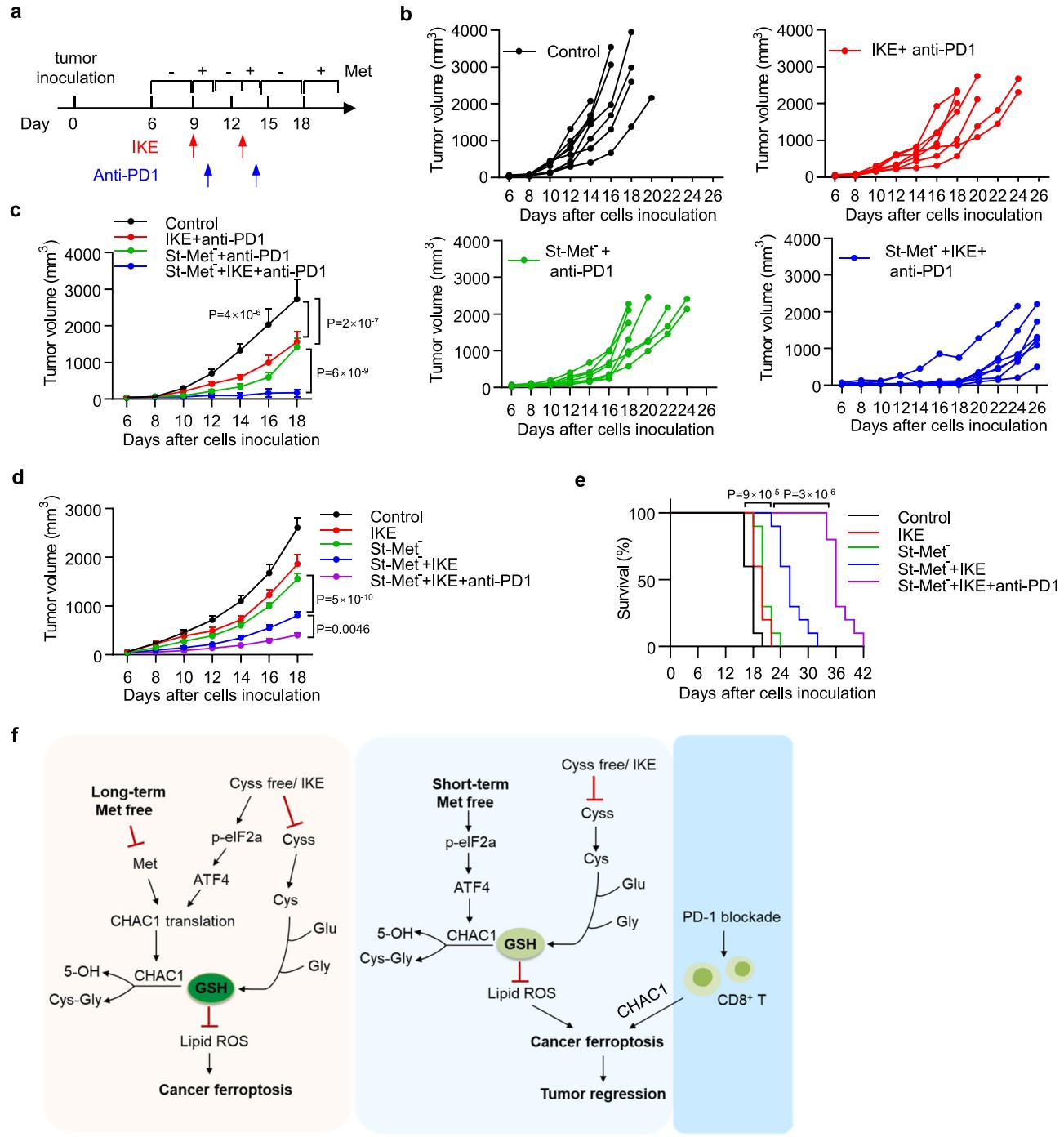

**Fig. 8 | Triple combination of dietary methionine intermittent starvation, IKE, and PD-1 blockade shows superior antitumor activity. a–c** Experimental design of B16F10 tumor in C57BL/6 mice (**a**). Tumor-bearing mice were fed with Met⁻ diet intermittently (St-Met⁻) and treated with anti-PD-1 antibody (100 μg/mouse) alone or anti-PD-1 antibody plus IKE (20 mg/kg) at indicated days (**a**). Individual tumor growth was monitored over time (**b**). Tumor volumes of different treatment groups were plotted as mean ± s.e.m., $n = 7$ (IKE + anti-PD1, St-Met⁻ + anti-PD1, and St-Met⁻ + IKE + anti-PD1) or 8 (Control) mice per group and $P$ values are determined by two-way ANOVA (**c**). **d, e** B16F10 tumor-bearing mice were fed with Met⁺ diet and treated with IKE (20 mg/kg), or fed with Met⁻ diet intermittently (St-Met⁻) and treated with IKE (20 mg/kg) or IKE plus anti-PD-1 antibody (100 μg/mouse), $n = 10$ mice per group. Tumor volumes of different treatment groups were plotted as mean ± s.e.m., and $P$ values were determined by two-way ANOVA (**d**). Kaplan–Meier survival curves of these animals (**e**) and $P$ values are determined by log-rank test (**e**). **f** Schematic representation of the proposed model. Under long-term methionine deprivation, CHAC1 translation induced by cystine deprivation is blocked, resulting in reduced GSH degradation and impaired ferroptosis onset (left). Upon short-term methionine pre-starvation, CHAC1 transcription is stimulated, but translation is not inhibited, resulting in enhanced GSH degradation and ferroptosis sensitization. PD-1 blockade-activated CD8⁺ T cells synergize with dietary methionine intermittent deprivation to induce potent cancer cell ferroptosis and improved antitumor immunity (right). Source data are provided as a Source Data file.

antitumor activity to reduce tumor volume (Fig. 8d and Supplementary Fig. 8a) increased tumoral lipid peroxidation (Supplementary Fig. 8b) and improve animal survival (Fig. 8e). These data suggest a

potential strategy for cancer combinational therapy utilizing dietary methionine intervention, ferroptosis inducer, and checkpoint blockade.

## Discussion

Reduced GSH functions as the co-factor of GPX4 to relieve lipid peroxidation and thus protect cells from ferroptosis. Cystine deprivation or system $x_c^-$ inhibition is considered to induce ferroptosis mainly by blocking the de novo synthesis of GSH[12,52]. However, the intracellular GSH pool is maintained not only by GSH synthesis but also by GSH degradation[53,54]. We demonstrated here that GSH degradation mediated by CHAC1 also contributes to the onset of ferroptosis. CHAC1 has been known as an enzyme to degrade intracellular GSH into 5-OH and Cys–Gly[44,55], but its regulation on ferroptosis has not been well recognized. The basal level of CHAC1 expression is suppressed across most cancer cells that we tested. And we found that CHAC1 was rapidly upregulated during ferroptosis induced by cystine deprivation or system $x_c^-$ inhibitor, which is mediated at the transcriptional level through activation of the eIF2α/ATF4 pathway. Deprivation of certain amino acids, such as cystine or methionine, is able to activate ER stress via GCN2-eIF2α-ATF4 cascade[18,56–59]. As a transcription factor, ATF4 then stimulates the transcriptions of various downstream genes, including CHAC1[24,27,55,60]. In line with our findings, CHAC1 was reported to be induced by cystine starvation, promote GSH degradation and enhance cystine starvation-induced cell death in human triple-negative breast cancer cells[27]. We had shown that disrupted activation of ER stress could decrease CHAC1 expression and inhibit ferroptosis induced by cystine deprivation. We further manipulated CHAC1 expression in multiple tumor cell lines and found that CHAC1 knockout or knockdown was delayed, while CHAC1 overexpression accelerated GSH depletion and ferroptosis onset. Moreover, prolonged exogenous CHAC1 over-expression is able to directly induce ferroptosis in the absence of synthetic inducers. Hence, our data indicate that CHAC1-promoted GSH degradation is a previously unrecognized mechanism to ensure the onset of ferroptosis initiated by cystine deprivation or system $x_c^-$ inhibition (Fig. 8f).

Deprivation of extracellular methionine is recently shown to suppress ferroptosis in response to system $x_c^-$ inhibitor or cystine withdrawal, which is explained by the proliferation arrest caused by methionine deprivation without detailed molecular mechanism[19]. Interestingly, the inhibitory effect of protein synthesis inhibitor CHX on cystine deprivation-induced ferroptosis is also observed in the same study[19]. We find that prolonged methionine deprivation indeed blocks ferroptosis in response to cystine deprivation or system $x_c^-$ inhibition across various types of human and mouse tumor cells. We reveal that the synthesis of CHAC1 induced by cystine deprivation is blocked by prolonged methionine deprivation at the translational level, resulting in inadequate GSH depletion and impaired ferroptosis onset. Conversely, short-term methionine starvation is able to enhance ferroptosis by stimulating CHAC1 transcription but without interruption of protein synthesis. Our data reveals the intricate regulatory mechanism of methionine metabolism on GSH homeostasis and tumor ferroptosis.

Dietary methionine intervention has shown antitumor effects in multiple animal tumor models and some prospective human clinical trials[1,2,61,62]. However, due to the multifaceted functions of methionine metabolism, the mechanism driving the antitumor effects of methionine intervention is not fully uncovered. A methionine-restricted diet over a 2-week period is able to inhibit tumor progression and enhance the antitumor activity of chemotherapy 5-fluorouracil (5-FU) or radiation in a patient-derived xenograft and autochthonous mouse tumor models[8]. Metabolomics revealed that methionine restriction was able to reduce methionine-related metabolites in circulation within two days, and disrupted nucleotide metabolism and redox balance in tumors at the end point[8]. Here, we used a methionine-free diet with intermittent treatment. It alone resulted in significant tumor regression and enhanced the antitumor activity of ferroptosis inducer IKE in the mouse B16 tumor model. Meantime, dietary methionine intermittent deprivation synergized with IKE to reduce GSH levels and increase lipid peroxidation and the expressions of PTGS2 and CHAC1 expression in tumors. Collectively, our results reveal that dietary methionine intervention exerts its antitumor effect through ferroptosis initiation or at least sensitization. In accordance with our findings, ferroptosis was recently reported to be induced and play a role in the progression of methionine-choline deficient (MCD) diet-induced nonalcoholic steatohepatitis (NASH)[63,64], although the underlying mechanism of MCD-induced hepatic ferroptosis is not clear. In addition, our findings suggest that it is important to identify optimal treatment regimens of methionine intervention for cancer treatment in future human clinical trials. From the perspective of ferroptosis, we find that intermittent but not sustained methionine deprivation is beneficial for ferroptosis induction and execution. This is conceptually similar to intermittent fasting (alternate-day or 5:2 intermittent fasting)[65], which has been shown to reduce tumor progression and sensitize tumor to chemotherapy as well as immunotherapy of PD-1 blockade in mouse models[66–69]. Fasting-mimicking diet elicits systemic and cellular metabolic switching, including changed glucose or lipid metabolism, mitochondrial biogenesis and protein synthesis, as well as signaling transduction through the insulin and growth hormone receptors[66,70–73]. Our data suggests that short-term methionine starvation may achieve a similar purpose to reprogramming metabolism and gene expressions in tumor cells, meanwhile, it avoids the undesired toxicity caused by long-term methionine deprivation.

Dietary methionine intermittent deprivation also synergizes with PD-1 blockade to inhibit tumor growth and enhance T cell-mediated immune response in mouse melanoma models. Although future in-depth studies may be worth to fully understand the role of methionine intermittent deprivation on tumor metabolism and antitumor immunity, we find here that short-term methionine starvation can cooperate with CD8$^+$ T cells to induce tumoral ferroptosis by converging on CHAC1. CHAC1 over-expression sensitizes the tumor to CTL-mediated cytotoxicity, whereas CHAC1 loss causes CTL resistance in vitro and immunotherapy resistance in vivo. Clinically, tumor CHAC1 correlates with clinical benefits and improved survival in cancer patients treated with checkpoint blockades. We thus propose that CHAC1-regulated tumoral ferroptosis contributes to the synergistic therapeutic efficacy of methionine intermittent deprivation in combination with PD-1 blockade. As an immunogenic cell death, ferroptotic cells are shown to release danger-associated molecular patterns to promote dendritic cell maturation and activate T cell-mediated adaptive immunity[33,74,75], which may also explain why tumoral ferroptosis-targeting therapy can cooperate with cancer immunity in vivo. Induction of tumor ferroptosis by depletion of extracellular cystine and cysteine using cyst(e)inase has been shown to synergize with checkpoint blockade to inhibit tumor progression and enhance anti-tumor T cell response[31,76]. Supplementation of arachidonic acid in combination with checkpoint blockade is also able to enhance tumor ferroptosis and anti-tumor immunity via reprogramming ACSL4-mediated lipid metabolism. Depletion of ACSL4 in tumor cells blocks ferroptosis initiation, suppresses T cell response, and accelerates tumor progression[30]. Furthermore, numerous nanomaterial-based ferroptosis inducers are developed and utilized for sensitizing ferroptosis and enhancing the therapeutic effect of immunotherapy across different tumor models[47,48,77–82]. While our work presents an alternative safe and feasible way to sensitize tumor cells to ferroptosis via dietary methionine intervention without the utilization of synthetic agents. Intermittent methionine deprivation alone can inhibit tumor growth and further synergizes PD-1 blockade-mediated therapy and antitumor T cell immunity. It is worth noting that human and mouse effector T cells are also sensitive to prolonged methionine deprivation[83–86]. In our study, a

methionine-free diet is applied for short period before the administration of an anti-PD-1 antibody, and the increased number and improved effector functions of tumor-infiltrating CD8[+] T cells are observed. Therefore, this regimen may avoid the toxicity of methionine deprivation on activated T cells.

In summary, we have shown that CHAC1 induced by amino acid deprivation promotes GSH degradation, facilitates ferroptosis, and sensitizes tumor against T cell-mediated cytotoxicity. Our work demonstrated complicated crosstalk between methionine metabolism, tumor ferroptosis, and antitumor immune response. Dietary methionine intermittent deprivation, but not sustained deprivation, may be a feasible regimen with translational value for in vivo ferroptosis sensitization and immunotherapy combination for cancer therapy.

## Methods

### Ethics statement
All studies reported in this manuscript comply with the relevant ethics regulations. The animal studies were conducted in accordance with the Institutional Animal Care and Use Committees and Institutional Review Board (IRB) of the School of Basic Medicine, Tongji Medical College, Huazhong University of Science and Technology.

### Reagents
L-Cystine, L-Methionine, and L-Glutamine were purchased from Sigma-Aldrich. Imidazole ketone erastin (IKE), Glutathione (GSH), and Cycloheximide (CHX) were purchased from Selleck Chemical. Erastin, RSL3, and Ferrostatin-1 were purchased from Cayman Chemical. Actinomycin D and GCN2iB were from MedChemExpress. BODIPY 581/591 C11, 2-Mercaptoethanol was purchased from Thermo Fisher Scientific.

### Cell culture
Human fibrosarcoma cell line HT-1080 (CCL-121), HEK293T (CRL-11268), and mouse melanoma cell line B16F10 (CRL-6475) were from the American Type Culture Collection (ATCC). Human hepatocarcinoma cell lines Hep3B (SCSP-5045), SNU387 (SCSP-5046) and SNU182 (SCSP-5047), Human renal carcinoma cell lines Caki-1 (SCSP-5064) and ACHN (SCSP-5063), mouse hepatocarcinoma cell line Hepa1−6 (SCSP-512) and mouse embryonic fibroblast cell line NIH/3T3 (SCSP-515) were from the China Center for Type Culture Collection (CCTCC). OS-RC-2 and 786-O cell lines are gifts from Dr. Xiangping Yang (Huazhong University of Science and Technology), who obtained them from CCTCC and ATCC. HT-29 originating from ATCC (HTB-38) is kindly provided by Dr. Zheng Wang (Huazhong University of Science and Technology), and PancO2 is a gift from Dr. Jun Zhao (Huazhong University of Science and Technology) obtained from the Cell Resource Center, Peking Union Medical College. HT-1080, Hep3B, and ACHN were cultured in MEM medium (Gibco, Thermo Fisher), Caki-1 was cultured in McCoy's 5 A medium (Procell), NIH3T3 and PancO2 were cultured in DMEM medium (Gibco, ThermoFisher Scientific), and all other cells were cultured in RPMI medium (Gibco, ThermoFisher Scientific). All culture mediums were supplemented with 10% FBS (Cegrogen Biotech). All cell lines were routinely tested for the presence of mycoplasma by a PCR-based method.

For cystine/methionine deprivation assays, culture media was prepared from no glutamine, no methionine, no cystine DMEM (Gibco, ThermoFisher Scientific) supplemented with 5% dialyzed serum (Cegrogen Biotech), and glutamine at a final concentration of 4 mM. This medium was used for the treatment of cystine and methionine double deprivation. For cystine or methionine single deprivation, either L-methionine or L-cystine was added to the above medium at a final concentration of 200 μM or 150 μM, respectively. As a control medium, both L-methionine and L-cystine were added. For short-time methionine starvation, cells were first maintained in the methionine-free medium for 4−8 h and then change to cystine-free or control medium for continuous culture.

### Lipid peroxidation, cell death, and immune profiling by flow cytometry
Tumor cells were seeded into 24-well plates at a density of $3 \times 10^4$ per well and treated with ferroptosis inducers for the indicated times. Cells were then harvested by trypsinization, centrifugation, and resuspended in 300 μl phosphate-buffered saline (PBS) containing 5 μM BODIPY 581/591 C11. Cells were incubated for 20 min at 37 °C in a tissue culture incubator, washed and resuspended in 200 μl PBS, then analyzed immediately on a flow cytometer (BD FACSVerse). The mean fluorescence intensities (MFI) from the PE channel (reduced C11) and FITC channel (oxidized C11) were both monitored, and their ratio was calculated for each sample. The data were normalized to the control sample, as shown by the relative lipid ROS.

For cell death assays, tumor cells ($2−4 \times 10^4$ cells/well) were seeded in 24-well plates and treated with different reagents or media-deficient specific amino acids. After incubation for the indicated time, cells, including the suspended dying cells, were collected and resuspended in 200 μl PBS containing 1 μg/ml Propidium Iodide (PI) or 7-Aminoactinomycin D (7-AAD) for 10 min and immediately run on a flow cytometer. PI or 7-AAD positive population was shown as the percentage of dead cells.

To quantify T cells and their cytokine production from tumor-infiltrating lymphocytes (TILs), fresh tumor tissues were harvested, excised, and ground into a single cell suspension by using a cell strainer (70 μm). TILs were obtained by density gradient centrifugation with lymphocyte separation medium (MP Biomedicals) and stimulated in culture medium supplemented with PMA (10 ng/ml), Ionomycin (1 μg/ml), Brefeldin A (1: 1000) and Monensin (1: 1000) at 37 °C for 5 h. Cells were first stained in PBS containing Zombie NIR Dye (Biolegend) to identify dead cells. Anti-CD45 (30-F11), anti-CD3 (145-2C11), anti-CD90 (53-2.1), anti-CD4 (GK1.5), and anti-CD8 (53-6.7) were then added for surface staining. Then cells were washed and resuspended in 500 μl Fixation/perm solution (Biolegend) at room temperature (RT) for 30 min, followed by staining with anti-IFNγ (XMG1.2) and anti-TNFα (MP6-XT22) for 20 min after washed with the Perm/Wash buffer (Biolegend). The cells were then washed and fixed in 4% formaldehyde (Biosharp). All samples were read on a flow cytometer and analyzed with FlowJo software (Version 10).

### Co-culture OVA[+] Luc[+] tumor cells with OT-I cells
Tumor cells were infected with lentivirus-expressing ovalbumin (OVA[+]) and nano-luciferase (Luc[+]), which are co-translated and separated by a self-cleaving 2A peptide. Infected cells were then selected with blasticidin for 10 days, and the activity of luciferase was quantified to ensure a similar amount of ovalbumin was expressed across different groups of cells.

The splenocytes from OT-I mice, C57BL/6-Tg (TcraTcrb) 1100Mjb/J, were isolated and suspended in 2 ml Red Blood Cell Lysis Buffer (Thermo Fisher Scientific) for 1 min. The cells were pelleted and resuspended at $2 \times 10^6$ cells/ml in RPMI culture medium containing 2.5 μg/ml OVA$_{257−264}$ peptide (GenScript), 10 ng/ml of mouse recombinant IL-2, and 40 μM 2-mercaptoethanol. The cells were incubated at 37 °C for 4 days. The amplified OT-I cells were then purified using EasySep™ mouse CD8[+] T Cell Isolation Kit (Stemcell).

OVA[+] Luc[+] tumor cells were seeded overnight, and then activated OT-I cells were added for co-culture. For quantification of luciferase activity, the supernatant was removed after co-culture, and the residual cells were lysed, and the total luciferase activity was assessed using the Nano-Glo Luciferase Assay kit (Promega) according to the manufacturer's instructions. For the quantification of dead tumor cells, 1 μM CFSE was used to label tumor cells. After co-culture, all cells were harvested by trypsinization and stained with 7-AAD. Tumor cell death

was analyzed by gating CFSE$^+$ 7-AAD$^+$ cell population on a flow cytometer. For quantification of lipid peroxidation, the supernatant was removed from the co-culture system, and the cells were harvested by trypsinization and stained with Pacific Blue anti-CD45 (30-F11) for 10 min at room temperature. And then, cells were resuspended in PBS containing 5 µM BODIPY 581/591 C11 and incubated for 15 min at 37 °C. Cells were washed and analyzed immediately with a flow cytometer.

## Quantitative PCR analysis

Total RNA was isolated from cultured cells or frozen tissues with RNAiso plus (TaKaRa). Then 500 ng total RNA was converted to cDNA using primeScript RT Master Mix (TaKaRa) with poly-dT and random hexamer primers. Quantitative PCR (qPCR) reactions were conducted using PowerUp SYBR Green Master Mix (Thermo Fisher Scientific) on a QuantStudio® 3 Real-Time PCR System (Thermo Fisher Scientific). The following primers were used for gene expression quantification: human GAPDH forward: TGGTATCGTGGAAGGACTC, human GAPDH reverse: AGTAGAGGCAGGGATGATG; human CHAC1 forward: GCTGT GGATTTTCGGGTACG, human CHAC1 reverse: CACACGGCCAGGC ATCTT; human CHAC1-Flag forward: TGGCGCTGGTGTATCTTCG, human CHAC1-Flag reverse: TGTCATCGTCATCCTTGTAGTCG; human ATF4 forward: GTGTTGGTGGGGGACTTGAT, human ATF4 reverse: GGAGAAGGCATCCTCCTTGC; human GCN2 forward: CAGCGACATAC TGAAGGGCA, human GCN2 reverse: AGGGATCCGCAGGTCAAAAG; mouse CHAC1 forward: AGTGTGGAAGCCGGACTTTG, mouse CHAC1 reverse: CACTCGGCCAGGCATCTTGT; mouse PTGS2 forward: TGAGT ACCGCAAACGCTTCT, mouse PTGS2 reverse: CAGCCATTTCCTTCT CTCCTGT; mouse HMOX1 forward: GCCTCCAGAGTTTCCGCATA; mouse HMOX1 reverse: AGGAAGCCATCACCAGCTTAAA; mouse TFRC forward: GGTTATGAGGAACCAGACCGT; mouse TFRC reverse: TGTTCCCACACTGGACTTCG; mouse ACTB forward: AGATCAAG ATCATTGCTCCTCCT, mouse ACTB reverse: ACGCAGCTCAGTAACA GTCC; qPCR products were confirmed as a single specific band by gel electrophoresis. QuantStudio Design and Analysis (v1.5.1) software was used for data analysis. The threshold cycle (Ct) values for each gene were normalized according to the endogenous control (GAPDH or ACTB). And the 2 − ∆∆Ct method was used for analyzing fold changes in mRNA expression.

## Immunoblotting

Cells were collected and lysed in RIPA Buffer (Thermo Fisher) containing protease inhibitor cocktail (MedChemExpress) and phosphatase inhibitors cocktail (MedChemExpress) on ice for 15 min. Insoluble material was removed by centrifugation at $13,000 \times g$ for 15 min. Protein concentration in supernatant was quantified by a BCA Protein Assay Kit (Thermo Fisher). Totally, 20–40 µg total protein was denatured at 95 °C in sample buffer (Thermo Fisher) for 10 min. Samples were separated via electrophoresis using SDS-PAGE gels (Biosharp) and transferred onto a 0.22 µm PVDF membrane (Millipore). Membranes were then blocked with 5% w/v nonfat dry milk or 3% BSA in TBS containing 0.1% Tween-20 for 1 h at RT, followed by overnight incubation with specific primary antibodies at 4 °C and HRP-conjugated secondary antibodies (Santa Cruz) for 2 h at RT. A luminescent signal was detected with Chemiluminescent HRP Substrate (Millipore) under ChemiDoc Imaging System (Tanon-5200Multi) and analyzed using Tanon AllDox-X software. The following primary antibodies were used: anti-CHAC1 (Proteintech, 15207, 1:1000), anti-ATF4 (Proteintech, 10835, 1:500), anti-Flag (Sigma,1804, 1:2000), anti-human eIF2α (CST, 5324, 1:2000), anti-EIF2S1 (phospho S51) (Abcam, ab32157, 1:1000), anti-GAPDH (Proteintech, 60004, 1:5000) and anti-alpha Tubulin (Proteintech, 11224, 1:5000).

## Generation of knockout and knockdown cells

Specific gene knockout tumor cells were generated with CRISPR–Cas9 technology. Single guide RNA (sgRNA) or scramble gRNA was synthesized and cloned into the lentiCRISPR v2 vector (Addgene, 52961). Then the plasmid was transfected into HEK293T cells together with psPAX2 and pMD2.G plasmids using Lipofectamine 2000 (Thermo Fisher Scientific). The culture medium was changed 12 h after transfection, and the supernatant containing the virus was collected after 72 h. Tumor cells were infected with lentivirus expressing each sgRNA, selected with puromycin for 3–4 days, and the efficiency of knockout was validated by immunoblotting. For a generation of CHAC1 knockout B16F10 cells, single-cell clones were selected and expanded. Multiple validated CHAC1 deficient clones were pooled for the experiments. Guide RNA sequences to target human CHAC1 was: CCAGTACAGCCGCTTTTCTC; Guide RNAs to target mouse CHAC1 were: TCGTTCGTGGCTATAGCCGA or ATAGCCGACGGTTCTGGCA.

Gene knockdown was achieved by the delivery of short hairpin RNAs (shRNAs) into tumor cells. shRNA oligonucleotide was cloned into pLKO.1 vector (Addgene, 10878), and similar to the above, lentivirus was produced and collected. Tumor cells were infected with lentivirus, and the positive cells were screened by puromycin for 3–5 days, and knockdown efficiency was validated by qPCR or immunoblotting. The following shRNAs targeting each gene were used in the study. Human shCHAC1-1: CCGGGACTTACTACTTGAAACTTTACTCGA GTAAAGTTTCAAGTAGTAAGTCTTTTTG, human shCHAC1-2: CCGGGA AGTACCTGAATGTGCGAGACTCGAGTCTCGCACATTCAGGTACTTCT TTTTTG, human shCHAC1-3: CCGGGAAGTACCTGAATGTGCGAGACT CGAGTCTCGCACATTCAGGTACTTCTTTTTTG, human shATF4: CCGG GCCAAGCACTTCAAACCTCATCTCGAGATGAGGTTTGAAGTGCTTGG CTTTTTG.human shGCN2:CCGGGCCTAACTGGTGAAGAAGTATCTCG AGATACTTCTTCACCAGTTAGGCTTTTTG.

## Cloning of human and mouse CHAC1

RNA from human or mouse tumor cells was reverse-transcribed. The cDNA of human or mouse CHAC1 was amplified by PCR using the following primers: human CHAC1 forward: ATGAAGCAGGAGTCTGCA GCCCCG, human CHAC1 reverse: CACCAGCGCCAGAGCCTGCTCGGT; mouse CHAC1 forward: ATGAAGCAGGAGTCCGCATCC; mouse CHAC1 reverse: GGTCAGTGCCAGAGGCTGCTC. The PCR was performed with the Phanta Max Super-Fidelity DNA Polymerase Kit (Vazyme) in the following conditions: for human CHAC1, 95 °C for 3 min, followed by 35 cycles of 95 °C for 15 s, 67 °C for 15 s and 72 °C for 2.5 min; for mouse CHAC1, 95 °C for 3 min, followed by 35 cycles of 95 °C for 15 s, 63 °C for 15 s, 72 °C for 2.5 min. The PCR products were cloned into an expression plasmid pLenti-CMV-3 x Flag using ClonExpress II One Step Cloning Kit (Vazyme). The plasmid DNAs were validated by Sanger sequencing and purified with an EndoFree Plasmid Midi Kit (CWBio) for transfection experiments. To construct Tet-On inducible CHAC1-Flag expressing plasmid, the above CHAC1 CDS cassette was cloned into pLV-Puro-TRE3G-3xFlag plasmid and validated by Sanger sequencing.

For transient transfection, HT-1080 cells were seeded into a 24-well plate at a density of $6 \times 10^4$ cells/well. The next day cells were transfected with different amounts of CHAC1-Flag expressing plasmid or an empty control plasmid using TurboFect Transfection Reagent (Thermo Fisher). After incubation for the indicated time, cells were collected, stained with PI, and analyzed by flow cytometry.

## Generation of Tet-On inducible CHAC1 expressing cells

A Tet-On 3G system was used to create the doxycycline-inducible CHAC1-expressing cells. B16F10 cells were first infected with a lentivector expressing Tet3G transactivator and then selected by Hygromycin B (300 µg/ml) for 10 days to generate Tet3G stable cell lines. These cells were then infected with the CHAC1 inducible lentivector pLV-Puro-TRE3G-CHAC1-3xFlag and selected by puromycin (2 µg/ml) for 3 days to create the double-stable Tet-On 3G CHAC1 inducible cell line. The inducible promoter TRE3G provides for very low basal expression and responses to the binding of Tet3G. Cells were cultured

in the medium containing 10% doxycycline-free serum. Upon supplementation with 0.5–1 μg/ml doxycycline, Tet3G binds specifically to the TRE3G promotor and activates transcription of the downstream CHAC1. Flag-tagged CHAC1 expression in these cells after 24–48 h of doxycycline treatment was confirmed by immunoblot.

## Glutathione quantification

Total GSH and GSSG concentrations in cell lysates were assessed using GSH/GSSG-Glo™ Assay (Promega, V6611) according to the manufacturer's instructions. Briefly, cells (5000 cells/well) were seeded into 96-well plates and next day, treated with ferroptosis inducers in the presence of an amino acid deficient medium. After appropriate incubation time, the culture medium was carefully removed, and cells were lysed with Total Glutathione Lysis Reagent or Oxidized Glutathione Lysis Reagent, followed by shaking at RT for 5 min. Luciferin Generation Reagent was then added and incubated for 30 min. Lastly, Luciferin Detection Reagent was added and mixed. After 15 min, the luminescence signal was measured on a microplate reader (BioTek Synergy H1). Standard curves of GSH and GSSG were generated along with samples and used for calculation. Subtraction of the GSSG reaction signal from the total Glutathione signal yielded the value of reduced GSH in the sample. In the meantime, another set of treated cells was used for cell number normalization by quantifying total protein concentration (BCA Protein Assay Kit).

Total GSH content in tumor tissues was quantified by a colorimetric-based assay using GSH and GSSG Assay Kit (Beyotime, S0053). Briefly, tumor tissues were frozen in liquid nitrogen and stored at −80 °C. After returning to RT, 2 mm (diameter) steel beads and solution (100 μl per 10 mg of tissue) were added to a 1.5 ml centrifuge tube containing tumor tissue. And then, tissues were fully homogenized by a tissue homogenizer (Jingxin, JXFSTPRP-24). The homogenate was centrifuged at $10{,}000 \times g$ at 4 °C for 10 min, and 10 μl of supernatant was mixed with 150 μl of total glutathione detection working solution. After incubation for 5 min, 50 μl NADPH solution was added, and absorbance at 532 nm was detected. The total GSH content in the samples was calculated by using a GSH standard curve.

## Malondialdehyde (MDA) assay

The MDA content in tumor tissue lysates was measured using the Lipid Peroxidation MDA Assay Kit (Biosharp, BL904A) based on the reaction between MDA and thiobarbituric acid (TBA). Tumor tissues were homogenized using a tissue homogenizer, and the supernatant was collected by centrifugation at $12{,}000 \times g$ for 10 min. Then 100 μl supernatant and 200 μl MDA detection working solution were mixed in a centrifuge tube. After being heated at 100 °C for 15 min, the supernatant was collected by centrifugation at $1000 \times g$ at RT for 10 min. Totally, 200 μl of the reaction mix was transferred to a 96-well plate, and the absorbance at 532 nm was measured. The MDA standard curve was generated along the samples. Meantime, the protein concentration of supernatant from tissue lysate was determined after further dilution. The MDA concentration of each sample was calculated using the standard curve and normalized to the protein concentration.

## NAD/NADH assay

The NAD/NADH levels in tumor tissue lysates were measured using NAD/NADH assay kit (Beyotime, S0175). Tumor tissues were homogenized in NAD/NADH extraction buffer, and the supernatant was collected by centrifugation at $12{,}000 \times g$ for 10 min. Totally, 50 μl of each sample was heated in a water bath at 60 °C to decompose NAD in order to detect NADH content. Then 20 μl supernatant with or without heat treatment was mixed with 90 μl alcohol dehydrogenase working solution. After incubation for 30 min in the dark at 37 °C, 10 μl chromogenic solution was added to each well. The absorbance of the samples at 450 nm was measured following incubation at 37 °C

for 30 min in the dark. The levels of NAD/NADH were calculated based on the standard curve and normalized to the protein concentration.

## Immunohistochemical (IHC) staining of 4HNE

Fresh mouse tumor tissues were fixed in 4% paraformaldehyde and embedded in paraffin. Deparaffinized sections were stained using IHC Prep & Detect kit (Proteintech, PK10019) according to the manufacturer's instructions. The sections were heated for antigen retrieval for 15 min and blocked with a blocking buffer for 1 h at RT. Then the sections were incubated with anti-4HNE (JaICA, HNEJ-2, 1:200) at 4 °C overnight. Following incubation with HRP-conjugated secondary antibody for 1 h, the sections were colored with diaminobenzidine (DAB) and redyed with the counter staining reagent. After dehydration and sealing treatment, the sections were scanned to collect bright field images by a slide-scanning platform (Pannoramic SCAN, 3DHISTECH). The images were analyzed using CaseViewer2.4 software (3DHISTECH).

## Animal experiments

Wild-type C57BL/6 mice, aged 6–8 weeks, were obtained from Beijing Vital River Laboratory Animal Technology Co., Ltd. C57BL/6-Tg (TcraTcrb) 1100Mjb/J (OT-1 mice) were kindly provided by Dr. Ning Wu (Huazhong University of Science and Technology), and mice were bred and housed in the animal facility of Huazhong University of Science and Technology. Both female and male OT-I mice aged 8–10 weeks were used in the experiments. All animals were housed under specific pathogen-free conditions in groups of 5 mice per cage and maintained in a humidity-controlled environment with a 12 h light/dark cycle at a temperature of 22–25 °C. Animal studies were conducted in accordance with the Institutional Animal Care and Use Committees and Institutional Review Board of the School of Basic Medicine, Tongji Medical College, Huazhong University of Science and Technology.

For the Hepa1–6 tumor model, $3 \times 10^6$ cells in Matrigel (Corning, 356234) were subcutaneously injected into the armpit of male C57BL/6 mice. In the sustained methionine deprivation experiment, the mice were fed either a 0% methionine (Met⁻) or 0.3% methionine (control) diet (XieTong biology, Jiangsu) until the end of the experiment. IKE (40 mg/kg) was administered intraperitoneally every other day, and liproxstatin-1 (10 mg/kg) was administrated every day. Animals were randomized into different groups after tumor cell inoculation. Animal body weight and subcutaneous tumor volume were measured using a balance and calipers, respectively. Tumor volume was calculated as length × width² × 0.5. Mice were euthanized when tumor size exceeded 2 cm in any direction, which was permitted by our institutional review board. We confirm that none of the mice included in this experiment exceeded this limit.

For B16F10 tumor model, $3 \times 10^5$ cells in 100 μl PBS were subcutaneously injected into the right flank of female C57BL/6 mice. To test the synergistic effect of dietary methionine intermittent starvation with either IKE or PD-1 blockade, the tumor-bearing mice were switched to a Met⁻ or control diet at day 3 after tumor inoculation, re-supplemented methionine (300 mg/kg) and IKE (40 mg/kg), 100 μg anti-PD-1 antibody (Bio X cell) or liproxstatin-1 (10 mg/kg) were intraperitoneally injected on indicated days. For the CHAC1 deficient B16 tumor model, 100 μg anti-PD-1 antibody was started on day 6 to test its combined effect with methionine intermittent starvation or on day 3 for its monotherapy. In the triple combination of dietary methionine intermittent starvation, tumor-bearing animals were treated with a methionine-free diet, 100 μg anti-PD-1 antibody, and a lower dose of IKE (20 mg/kg) periodically. Animals were randomized into different groups after tumor cells inoculation, and at least seven mice were used for each group, unless otherwise indicated. Tumor volume was monitored every other day and was calculated as length × width² × 0.5. Mice

were euthanized when tumor size exceeded 2 cm in any direction or when mice reached a humane endpoint, such as rapid weight loss, hunched back posture, or difficulty in feeding. For the survival analysis, B16F10 tumor-bearing mice were considered to have reached the survival endpoint when their tumor volume exceeded 2000 mm³ or mice reached the humane endpoint. For analysis of tumor growth and ferroptosis or immune-related parameters in corresponding tumor tissues, at least five tumors per group were kept until the end of experiments for sufficient statistical analysis. Subcutaneous tumor tissues were then surgically removed. Tumor weights, T-cell infiltrations, MDA contents, GSH contents, or mRNA levels of ferroptosis-related genes in these tumor tissues were quantified.

### Methionine quantification

The methionine concentration in mouse serum was measured using Methionine Assay Kit (Abcam, ab234041). Tumor-bearing mice were fed with Met⁻ or control diet and re-supplemented with methionine at indicated days. Blood samples were obtained through tail bleeding, stored at 4 °C overnight, and centrifuged at $1200 \times g$ for 20 min at RT. The serum was collected and diluted to 100 µl with PBS, and then 2 µl of Sample Clean-up mix was added and incubated at 37 °C for 30 min. Samples were filtered through a 10 kDa spin column ($10,000 \times g$, 4 °C, 10 min), and the ultrafiltrate was retained. For each sample, two parallel wells were prepared, one for the determination of methionine and one for a sample background control. Totally, 5 µl ultrafiltrate and 20 µl detection buffer were added to each well. Then 25 µl reaction mix and 25 µl background control mix were added to their parallel sample wells, respectively. Meanwhile, 100 µM methionine standard was used to prepare the standard curve. Fluorescence was read in endpoint mode (Ex/Em = 535/587 nm) after 30 min of incubation of the plate at 37 °C. The absolute methionine concentration was calculated for each sample using the standard curve.

### LC–MS quantification of metabolites

Tumor cells were seeded into 10 cm culture dishes and treated with amino acid deprivation for indicated times. The culture medium was quickly removed, and cells were washed with cold PBS, collected with a cell scraper, and incubated with 200 µl of 250 mM dithiothreitol (DTT) aqueous solution. Then 800 µl of pre-chilled methanol was added, and the mixture was incubated for 30 min at 4 °C and centrifuged at $13,000 \times g$ for 10 min at 4 °C. The supernatant was transferred to a clean tube and dried using SpeedVac. The dried extracts were redissolved for LC-MS analysis. The column ACQUITY UPLC HSS T3 (1.8 µm, 2.1 × 100 mm) was used for reversed-phase chromatographic analysis. Ultra-performance Liquid Chromatography (Agilent 1290 II, Agilent Technologies) coupled to Quadrupole-TOF MS (5600 Triple TOF Plus, AB SCIEX) was utilized to acquire metabolome data. All detected ions were extracted using MarkerView1.3 (AB Sciex, Concord). PeakView 2.2 (AB Sciex, Concord) was applied to extract MS/MS data and perform a comparison with the Metabolites database (AB Sciex, Concord), HMDB, METLIN, and standard references to annotate ion ID.

### Isotopic tracing analysis

HT-1080 cells were seeded into 6-well plates ($5 \times 10^5$ cells/ml, triplicate wells per condition) in complete medium overnight. Cells were moved to a medium containing 100 µm [¹³C2]-ʟ-Cystine (IsoReag, IR-30573) and [³⁴S]-L-Methionine (TRC/Toronto Research Chemicals, ZTR-M260443) and cultured for 24 h. Cells were then washed with PBS, and starved with the cystine-free medium containing [³⁴S]-ʟ-Methionine alone for 8 h. As a control, cells were maintained in [¹³C2]-ʟ-Cystine and [³⁴S]-ʟ-Methionine-containing medium until the end of the experiment. The medium was aspirated, and cells were washed with PBS and collected with ice-cold methanol plus a reducing substance quenching agent. Samples were injected and analyzed by an Agilent 7890 A gas chromatography system coupled to an Agilent 5975 C inert MSD system (Agilent Technologies Inc., CA, USA). The peak area of metabolites was calculated using phenylalanine-d5 as a reference.

### Statistical analysis

No statistical method was used to predetermine the sample size. Data were shown as individual values, means ± s.e.m. or means ± SD of the indicated numbers. Two-tailed *t*-tests or Mann–Whitney tests were used to compare two independent groups; ANOVA models were used to compare continuous outcomes across multiple experimental groups, and Tukey and Sidak corrections were used to adjust *P* values for multiple comparisons. Survival functions were estimated by Kaplan–Meier methods and the log-rank test was used to calculate statistical differences. Statistical analyses were performed using GraphPad Prism8 software (GraphPad Software, Inc.).

### Reporting summary

Further information on research design is available in the Nature Portfolio Reporting Summary linked to this article.

### Data availability

All data to support the conclusions in this paper can be found in the Article, Supplementary Information, or Source Data file, which are provided alongside this paper. Metabolomics data corresponding to Fig. 2a, b and Supplementary Fig. 2a generated in this study is provided in Supplementary Data 1. Genomic data corresponding to Fig. 3a is publicly available at the GEO with the accession number GSE60422. Genomic data corresponding to Fig. 7f, g and Supplementary Fig. 7b, c is publicly available at ENA under the accession number PRJEB23709 and Cancer-Immu platform (http://bioinfo.vanderbilt.edu/database/Cancer-Immu/) or directly through phs000452.v2.p1. Source data are provided in this paper.

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

## Acknowledgements

We thank Dr. Ning Wu and Dr. Wei Li for helpful discussions and advice for this paper. This work was supported in part by grants from the National Natural Science Foundation of China to W. Wang (grant 81902930 and 82073177) and F. Lu (grant 82102930), the National Key Research and Development Program of China to W. Wang (grant 2021YFC2400600/2021YFC2400602), the Natural Science Foundation of Hubei Province to W. Wang (grant 2022CFA056) and the Independent Innovation Grant from Huazhong University of Science & Technology to W. Wang (grant 2021GCRC072).

## Author contributions

Y.X., F.L., and W.W. conceived the idea, designed the experiments, and composed the paper. Y.X. and F.L. conducted experiments; Z.C., Y.G., and J.Z. assisted in animal experiments; Y.L. assisted in flow cytometry analysis; J.L and Y.L. assisted in qPCR; S.C., X.X.L., Y.Z., and Y.L. assisted in tissue cultures; Z.T. and X.L. assisted in bioinformatics analysis; X.C. and J.C. contributed to the interpretation of the results. W.W. supervised the project.

## Competing interests

The authors declare no competing interests.
