## [Peer Review File · Nature Communications]

This manuscript has been previously reviewed at another journal that is not operating a transparent peer review scheme. This document only contains reviewer comments and rebuttal letters for versions considered at *Nature Communications*.

REVIEWERS' COMMENTS

Reviewer #1 (Remarks to the Author):

This is a revised manuscript that was transferred from another Nature sister journal. In general, the authors have done a good job and addressed most of my comments appropriately. As such, I just have two more rather minor points left:

- The last paragraph of point 1 in the response letter might be not fully justified. The reason why cells with reduced proliferation are more resistant to ferroptosis are only in part related to lower cell densities as this is usually because of an increased generation of oxygen radicals due to higher metabolic rates.
- In the GSH supplementation experiments, it would have been more appropriate to use glutathione ethyl ester (GSH-EE) as it is a cell-permeable derivative of GSH and more readily available particularly in human cancer cell lines.

Reviewer #2 (Remarks to the Author):

The authors have now provided reasonable explanations on: (i) the control of ferroptosis by GSH, (ii) potential differences in the responses of different cell lines. (iii) transcriptional regulation of CHAC1, (iv) role of methionine vs cysteine contributions through flux tracing, and (v) more clearly elucidating the contributions of immune cell compartment. The connection between ferroptosis and immune cell response, which was initially tenuous given the lack of clarity, is now more logically conveyed as a result of new data. Given the extensive revision with new experimental data, this reviewer is satisfied with the revision.

* Reviewer #2 has also asked the editor to provide these comments to the authors regarding previous Reviewer #3's concerns:

In response to Reviewer 3's comments, the authors have: (i) clarified the novelty of CHAC1-regulated ferroptosis and CD8+ T cell-mediated immunotherapy, (ii) addressed why the tumor growth kinetics of cell lines may be different, (iii) corrected the error with the unit conversion, (iv) added new, longer survival curve data, (v) increased animal sample size for I/O treatment, and (v) revising or toning down certain claims. This reviewer appreciates that the authors performed rather substantial additional animal studies to address the comments of Reviewer 3. I also believed that Reviewer 3 found the study to be of significance and the concerns on the in vivo aspects have been sufficiently addressed.

Reviewer #4 (Remarks to the Author):

I have reviewed the current revised manuscript along with the response raised by the Reviewers.

The authors have done a tremendous amount of work (e.g. tracing studies) to clarify and address the reviewer comments both phenotypically and mechanistically, at least within the scope of the current manuscript. While I appreciate that there was some apprehension about the novelty of the study with respect to the ATF4-CHOP-CHAC1 angle, I do believe other aspects such as the discrimination between the biological effects of LT- vs ST-methionine restriction provides sufficiently new insight. In other words, the strengths outweigh some of the perceived weaknesses and that the overall results are noteworthy. The body of work is more than incremental vis a vis the established literature.

I should also point out that this general area, methionine, ferroptosis, and immunity is a highly active area of investigation so the timing is particularly significant.

Given the breadth of cell lines tested, there will be immense interest in these findings from the field. Some of the claims are slightly far-reaching, as pointed out by the Reviewer, and the authors have

tempered their conclusions as prescribed.

In terms of the immunological links, my opinion is that this manuscript primarily deals with the role of methionine on tumor intrinsic metabolism/cell death (e.g. ferroptosis). Any insights on immunity are tumor extrinsic and out-of-scope to require additional experiment work to define how methionine regulates T cell responses. While desirable, this is a tall order and in my view, would ascend the manuscript to a different tier of journal.

RESPONSE TO REVIEWERS' COMMENTS

Reviewer #1 (Remarks to the Author):

This is a revised manuscript that was transferred from another Nature sister journal. In general, the authors have done a good job and addressed most of my comments appropriately. As such, I just have two more rather minor points left:

Response: We appreciate the Reviewer #1's positive comments.

1. The last paragraph of point 1 in the response letter might be not fully justified. The reason why cells with reduced proliferation are more resistant to ferroptosis are only in part related to lower cell densities as this is usually because of an increased generation of oxygen radicals due to higher metabolic rates.

Response: We might misunderstand previously on the point 1 raised by the reviewer and addressed it based on the regulation of cell density on ferroptosis sensitivity. Here, we understand that the reviewer wants us to test whether long-term methionine deprivation-mediated ferroptosis resistance observed in our cells (Figure 1) is caused by the reduced cell proliferation, since slowly proliferated cells would have the lower metabolic rates and generate less oxygen radicals. We thus performed the following experiments to further address this point.

1) We monitored the dynamic changes of total ROS and lipid ROS in HT-1080 cells in response to methionine deprivation. We found that the cell proliferation was indeed reduced after 16 hours of methionine deprivation (Figure 1a), and the total ROS and lipid ROS levels were both increased at first and then gradually decreased. But at the end the levels of total ROS and lipid ROS were still comparable with control group (Response Figure 1a, b).

2) We further quantified the total ROS and lipid ROS in HT-1080 cells treated with cystine and methionine co-deprivation. We found that both total ROS and lipid ROS levels were increased by cystine deprivation alone, and were then significantly attenuated by methionine co-deprivation. However, methionine deprivation alone did not reduce, even slightly increased the total ROS and lipid ROS (Response Figure 1c, d). These data suggest that long-term methionine deprivation-mediated ferroptosis resistance in our cells may not be caused by the reduced ROS production.

3) As a control, we also examined the effect of methionine deprivation on ROS production in CT26 cells, where long-term methionine deprivation enhanced cystine deprivation-induced ferroptosis (Response Figure 1e). We found that the cell proliferation quantified by the cell viability was reduced along with the time of methionine deprivation (Response Figure 1f), while the total ROS and lipid ROS levels were both increased gradually upon methionine deprivation

(Response Figure 1g, h). In CT26 cells treated with cystine and methionine co-deprivation, both total ROS and lipid ROS levels were increased by cystine deprivation alone, and were further enhanced by methionine co-deprivation (Response Figure 1i, j). These results suggest that although methionine deprivation reduces cell proliferation, it can still cause increased ROS production.

Response Figure 1

Response Figure 1. Methionine deprivation inhibits cell proliferation without reducing intracellular total ROS and lipid ROS. **a, b.** The relative total ROS (**a**) and lipid ROS level (**b**) in HT-1080 cells treated with methionine deprivation for indicated time. **c, d.** The relative total ROS (**c**) and lipid ROS level (**d**) in HT-1080 cells treated with methionine or cystine deprivation or their co-deprivation for 8 hours. **e.** Percentages of dead CT26 cells with cystine or methionine single deprivation or their co-deprivation. **f.** Cell viability of CT26 cells treated with methionine deprivation for indicated time. **g, h.** The relative total ROS (**g**) and lipid ROS level (**h**) in CT26 cells treated with methionine deprivation for indicated time. **i, j.** The relative total ROS (**i**) and lipid ROS level (**j**) in CT26 cells treated with methionine or cystine deprivation or their co-deprivation for 24 hours.

Therefore, our results suggest that methionine deprivation inhibits tumor cell proliferation without reducing the intracellular ROS level, and methionine

deprivation-mediated ferroptosis resistance may not simply due to the reduced cell proliferation rate.

2. In the GSH supplementation experiments, it would have been more appropriate to use glutathione ethyl ester (GSH-EE) as it is a cell-permeable derivative of GSH and more readily available particularly in human cancer cell lines.

Response: We previously showed that the supplementation of 200 μM exogenous GSH could partially recovery of intracellular GSH and fully rescue the cell death induced by cystine deprivation in HT-1080 cells (Figure 2k). Here, based on the reviewer's suggestion we have repeated the GSH rescue experiments using the glutathione ethyl ester (GSH-EE). We found that 50 μM exogenous GSH-EE was enough to rescue the intracellular GSH loss and the cell death of HT-1080 in response to cystine deprivation (Response Figure 2a, b), suggesting that GSH-EE is more effective than GSH for elevating the intracellular GSH level. Again, these results indicate that preventing GSH depletion from exceeding the death threshold is sufficient to block ferroptosis.

Response Figure 2

Response Figure 2. GSH-EE is more effective to rescue intracellular GSH loss and cell death induced by cystine deprivation. a, b. Total GSH content (a) and time points matched cell death (b) of HT-1080 cells treated with cystine deprivation plus exogenous GSH-EE.

Reviewer #2 (Remarks to the Author):

The authors have now provided reasonable explanations on: (i) the control of ferroptosis by GSH, (ii) potential differences in the responses of different cell lines. (iii) transcriptional regulation of CHAC1, (iv) role of methionine vs cysteine contributions through flux tracing, and (v) more clearly elucidating the contributions of immune cell compartment. The connection between ferroptosis and immune cell response, which was initially tenuous given the lack of clarity, is now more logically conveyed as a result of new data. Given

the extensive revision with new experimental data, this reviewer is satisfied with the revision.

* Reviewer #2 has also asked the editor to provide these comments to the authors regarding previous Reviewer #3's concerns:

In response to Reviewer 3's comments, the authors have: (i) clarified the novelty of CHAC1-regulated ferroptosis and CD8+ T cell-mediated immunotherapy, (ii) addressed why the tumor growth kinetics of cell lines may be different, (iii) corrected the error with the unit conversion, (iv) added new, longer survival curve data, (iv) increased animal sample size for I/O treatment, and (v) revising or toning down certain claims. This reviewer appreciates that the authors performed rather substantial additional animal studies to address the comments of Reviewer 3. I also believed that Reviewer 3 found the study to be of significance and the concerns on the in vivo aspects have been sufficiently addressed.

Response: We thank Reviewer #2 for his/her kind words and acknowledging our extensive revisions.

Reviewer #4 (Remarks to the Author):

I have reviewed the current revised manuscript along with the response raised by the Reviewers.

The authors have done a tremendous amount of work (e.g. tracing studies) to clarify and address the reviewer comments both phenotypically and mechanistically, at least within the scope of the current manuscript. While I appreciate that there was some apprehension about the novelty of the study with respect to the ATF4-CHOP-CHAC1 angle, I do believe other aspects such as the discrimination between the biological effects of LT- vs ST-methionine restriction provides sufficiently new insight. In other words, the strengths outweigh some of the perceived weaknesses and that the overall results are noteworthy. The body of work is more than incremental vis a vis the established literature.

I should also point out that this general area, methionine, ferroptosis, and immunity is a highly active area of investigation so the timing is particularly significant.

Given the breath of cell lines tested, there will be immense interest in these findings from the field. Some of the claims are slightly far-reaching, as pointed out by the Reviewer, and the authors have tempered their conclusions as prescribed.

In terms of the immunological links, my opinion is that this manuscript primary deals with the role of methionine on tumor intrinsic metabolism/cell death (e.g. ferroptosis). Any insights on immunity are tumor extrinsic and out-of-scope to require additional experiment work to define how methionine regulates T cell responses. While desirable, this is a tall order and in my view, would ascend the manuscript to a different tier of journal.

Response: We appreciate the reviewer's positive comments and the recognition for the importance and quality of our work.